# TabMGP: Martingale Posterior with TabPFN

**Kenyon Ng** [1]  **Edwin Fong** [2]  **David T. Frazier** [1]  **Jeremias Knoblauch** [3]  **Susan Wei** [1]

## Abstract

Bayesian inference provides principled uncertainty quantification but is often limited by the challenges of prior and likelihood elicitation. The martingale posterior (MGP) (Fong et al., 2023) offers an alternative by replacing these requirements with a predictive rule. In addition, the MGP focuses inference on parameters defined through a loss function. This framework is especially resonant in the era of foundation transformers; practitioners increasingly leverage models like TabPFN for their state-of-the-art capabilities, yet often require epistemic uncertainty for a scientific estimand $\theta$ that need not parameterise the implicit latent model. The MGP provides a mechanism to recover these posterior distributions. We introduce TabMGP, an MGP built on TabPFN for tabular data. TabMGP produces credible sets with near-nominal coverage and often outperforms both handcrafted MGP constructions and standard Bayesian baselines.

## 1. Introduction

Classical Bayesian inference provides a principled framework for uncertainty quantification, but its formulation hinges on the specification of a prior–likelihood pair. This requirement can be burdensome in practice. Priors are often placed on uninterpretable parameters, and "uninformative" defaults can dominate inference in data-scarce or high-dimensional regimes (Zhang et al., 2022). The problem is further compounded by the frequent misspecification of the likelihood in modern applications. Even when a model is settled upon, approximating the resulting posterior in high dimensions remains a formidable computational challenge.

---
[1]Department of Econometrics and Business Statistics, Monash University, Melbourne. [2]Department of Statistics and Actuarial Science, University of Hong Kong. [3]Department of Statistical Science, University College London.. Correspondence to: Susan Wei <susan.wei@monash.edu>.

*Proceedings of the 43rd International Conference on Machine Learning*, Seoul, South Korea. PMLR 306, 2026. Copyright 2026 by the author(s).

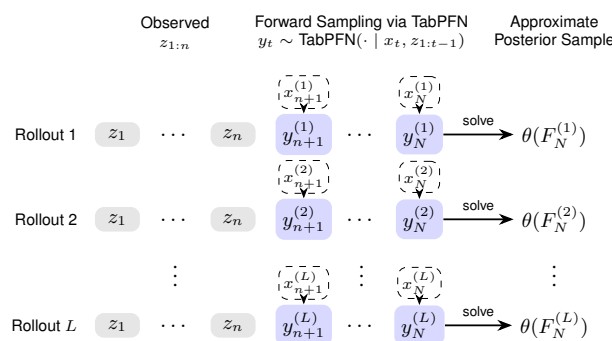

*Figure 1.* TabMGP for obtaining posterior samples of $\theta(F_\infty) \mid z_{1:n}$, where each $z = (x, y)$ represents a covariate-response pair. We perform **forward sampling** with TabPFN to generate $L$ independent continuations of the observed dataset $z_{1:n}$. Since TabPFN does not model covariates, the forward sampling of $x_i$ is performed via a separate process, while TabPFN provides the conditional response $y_i \sim \text{TabPFN}(\cdot \mid x_i, z_{1:i-1})$. At the end of each rollout $l$, we form the empirical measure $F_N^{(l)} = \frac{1}{N}\left(\sum_{i=1}^n \delta_{z_i} + \sum_{i=n+1}^N \delta_{z_i^{(l)}}\right)$ and collect $\theta(F_N^{(l)})$ as one **approximate posterior sample**.

These limitations have catalysed a wave of generalisations, including Bayesian predictive inference (BPI) (Fortini & Petrone, 2025), generalised Bayes (GB) (Bissiri et al., 2016), and optimisation-centric Bayes (see, e.g., Knoblauch et al., 2022; Wild et al., 2023; Shen et al., 2025), collectively marking the emergence of a **post-Bayesian paradigm.** Within this paradigm, BPI shifts the focus from prior and likelihood elicitation to the sequence of predictions themselves. Specifically, the analyst provides a **predictive rule**, defined as a sequence of one-step-ahead predictive distributions $(P_i)_{i \geq 0}$. BPI is post-Bayesian in the sense that the predictive rule need not correspond to a standard Bayesian posterior predictive distribution (PPD). Instead, the framework requires only that the sequence $(P_i)_{i \geq 0}$ converges weakly to a random limiting measure $F_\infty$ almost surely.

We can then ask for the posterior distribution of $F_\infty$ given data $z_{1:n}$. In this role, $F_\infty$ represents the **latent Bayesian model** internalised by the predictive rule, a state of knowledge that remains implicit but can be queried through simulation. Specifically, we may obtain posterior samples of $F_\infty \mid z_{1:n}$ through a process of **forward sampling**: starting from the observed dataset $z_{1:n}$, one repeatedly draws the next observation from the predictive rule conditional on the

data so far, appends it to the dataset, and continues. In practice, the forward sampling process is terminated at some large $N$ such that $F_N$ serves as an approximate posterior sample of $F_\infty \mid z_{1:n}$.

Complementary to BPI is GB, a framework that argues we can update a prior to a posterior distribution over parameters that are connected to observations through a loss function rather than a likelihood. This allows for significant flexibility as the object of interest $\theta$ is generally defined as the minimiser of an expected loss and need not index the underlying Bayesian model. Despite this flexibility, GB still requires the specification of an explicit prior.

The martingale posterior (MGP) emerges as a powerful synthesis of these directions, one that is especially resonant in the era of foundation transformers (Bommasani et al., 2022), which have recently begun to dominate tabular data tasks—a domain long considered a stronghold of traditional, non-deep-learning methods. Practitioners confronted with a new tabular dataset will increasingly turn to pretrained models like TabPFN (Hollmann et al., 2022) rather than design a bespoke likelihood–prior pair and navigate the complexities of Markov chain Monte Carlo or variational inference. For these users, the parameters of scientific interest are effectively decoupled from the internals of the transformer, necessitating a framework that can recover the posterior distribution of a functional $\theta(F_\infty)$ directly from the transformer's predictive outputs. The MGP provides this mechanism by treating the transformer as a predictive engine for the latent model $F_\infty$, while the GB loss function specifies the functional $\theta$ to be extracted.

The forward sampling mechanism above for $F_\infty \mid z_{1:n}$ provides the computational means for this recovery. The procedure remains the same except that, at termination, we report $\theta(F_N)$ as an approximate posterior sample from $\theta(F_\infty) \mid z_{1:n}$. This sampling scheme is well suited to transformers: **next-token prediction** *is* the predictive rule, and forward sampling mirrors *autoregressive generation*, where each token is sampled from the conditional distribution given the context, appended to the context, and the process is repeated. This procedure is visualised in Figure 1.

Yet, despite this alignment, the martingale posterior literature has focused almost exclusively on handcrafted predictive rules that satisfy exact martingale properties (e.g., Fong et al., 2023; Ghalebikesabi et al., 2023; Huk et al., 2024); see Section 4 for a review. These approaches are theoretically rigorous, but they require problem-specific design choices to tune the predictive rule for the dataset at hand. Moreover, the emphasis on exact martingale structure overstates this condition's importance. Although the martingale property is a sufficient condition to guarantee the existence of a limiting law $F_\infty$, it is **not necessary** for delivering epistemic uncertainty for $\theta(F_\infty)$. We argue that the

reliance on this restrictive sufficient condition has impeded the broader adoption of MGPs and prevents their application to high-capacity models.

This motivates our central question: can modern foundation transformers be used as predictive rules in martingale posteriors? We answer in the affirmative by introducing **TabMGP**, the first martingale posterior powered by a large pretrained foundation transformer. Specifically, we instantiate the predictive rule with **TabPFN** (Hollmann et al., 2022; 2025), a transformer trained on a broad range of synthetic tabular data that now achieves state-of-the-art results for tabular classification and regression tasks.

Our results show that TabMGP consistently provides reliable uncertainty quantification, often matching or surpassing both handcrafted martingale posteriors and classical Bayesian inference. These findings demonstrate that the MGP framework is considerably more robust than initially theorised. Even when high-capacity predictive rules like TabPFN do not strictly satisfy formal sufficient conditions such as the martingale property, the resulting posteriors remain empirically stable, providing superior frequentist coverage and tighter credible sets than standard baselines. By establishing a new method for the modern Bayesian toolkit, this work opens new avenues for broadening the MGP framework to the era of foundation models.

## 2. Martingale Posterior

The martingale posterior (Fong et al., 2023) is a recently proposed statistical inference method within the BPI framework (see Fortini & Petrone, 2025, for a review). We follow the notation of Fortini & Petrone (2025) in our discussion.

**Notation.** Let $(Z_i)_{i \geq 1}$ be random variables taking values in a space $\mathcal{Z}$. For any probability law $\mathbb{P}$ for the sequence $(Z_i)_{i \geq 1} \in \mathcal{Z}^\infty$, define the *predictive rule* as the sequence of predictive distributions $P_0(\cdot) \coloneqq \mathbb{P}(Z_1 \in \cdot)$ and, for $i \geq 1$,

$$P_i(\cdot) \coloneqq \mathbb{P}(Z_{i+1} \in \cdot \mid Z_{1:i}).$$

We write $P_i(\cdot \mid z_{1:i})$ for the realisation given $z_{1:i}$. We also write $\mathbb{P}$-a.s. for "with $\mathbb{P}$-probability one". We use $p_i$ to denote the density or mass function of $P_i$.

The martingale posterior requires two ingredients:

1. A predictive rule $(P_i)_{i \geq 0}$; and
2. A functional of interest $\theta : \mathcal{P}(\mathcal{Z}) \to \Theta \subseteq \mathbb{R}^p$, where $\mathcal{P}(\mathcal{Z})$ denotes the set of all distributions $F$ on $\mathcal{Z}$.

In this work, we focus on a risk minimiser that minimises a given loss function $\ell : \mathcal{Z} \times \Theta \to \mathbb{R}$:

$$\theta(F) \coloneqq \arg\min_{\vartheta \in \Theta} \int \ell(z, \vartheta) \, \mathrm{d}F(z). \tag{1}$$

For the martingale posterior distribution to be well-defined, the sequence of predictive distributions $(P_i)_{i \geq 0}$ must converge $\mathbb{P}$-a.s. to a random probability measure $F_\infty$, which represents the limiting empirical distribution of $(Z_i)_{i \geq 1}$. Here, $F_\infty$ is analogous to the statistical model in classical Bayes, and the probability laws of $F_\infty$ and $F_\infty \mid z_{1:n}$ are the prior and posterior, respectively. This posterior law of $F_\infty \mid z_{1:n}$ then induces a posterior distribution on functionals of $F_\infty$, which is called the martingale posterior.

**Definition.** Let $z_{1:n} \in \mathcal{Z}^n$ denote the observed data and $\Pi(F_\infty \mid z_{1:n})$ the posterior law of $F_\infty$ given $z_{1:n}$. The *martingale posterior* distribution is defined as the posterior law of $\theta(F_\infty)$:

$$\Pi_\infty(\theta(F_\infty) \in A \mid z_{1:n}) = \int \mathbf{1}(\theta(F_\infty) \in A) \, d\Pi(F_\infty \mid z_{1:n}),$$
(2)

for all measurable sets $A \subseteq \Theta$. The classical Bayesian posterior can be recovered from (2) as a special case by using a Bayesian PPD as the predictive rule and the corresponding negative log-likelihood as the loss function (Appendix B).

**Sufficient conditions.** A sufficient, but not necessary, condition for the existence of $F_\infty$ and hence for the martingale posterior distribution to be well-defined, is that the sequence $(P_i)_{i \geq 0}$ is a martingale, or equivalently, that the sequence $(Z_i)_{i \geq 1}$ is conditionally identically distributed (c.i.d.) (Berti et al., 2004). The martingale property stipulates that $(P_i)_{i \geq 0}$ satisfies, for every $i \geq 0$ and every measurable set $A$:

$$\mathbb{E}[P_{i+1}(A) \mid Z_{1:i}] = P_i(A).$$
(3)

Relaxations of this sufficient condition remain an active area of research (see, e.g., Battiston & Cappello, 2025). As noted in the introduction, the term "martingale posterior" is somewhat of a misnomer. Any predictive rule with a well-defined $F_\infty$ limit induces a posterior distribution as in (2). In this paper, we use the term broadly, so even predictive rules that violate the martingale property may still induce valid "martingale" posteriors.

**Computation.** The martingale posterior (2) generally does not admit a closed-form expression and is approximated by the *finite martingale posterior*. This is the posterior law of $\theta(F_N)$, where $F_N$ is the empirical distribution of $Z_{1:N}$. A draw from $\theta(F_N)$ is obtained by *forward sampling* a simulation rollout $Z_{n+1:N}$ from the predictive rule:

$$Z_{i+1} \sim P_i, \quad i = n, \ldots, N-1,$$

then forming $F_N$ and computing $\theta(F_N)$ (Fong et al., 2023). Repeating this produces $L$ samples $\{\theta^{(l)}\}_{l=1}^L$. These rollouts are independent and thus embarrassingly parallel.

## 3. TabMGP

Given the strong predictive performance of modern foundation models (Bommasani et al., 2022), it is natural to ask whether such models can serve as a predictive rule in the martingale posterior framework. We describe our proposed methodology, which employs TabPFN (Hollmann et al., 2022; 2025) as the predictive rule.

Given that TabPFN is designed for supervised prediction, we specialise the martingale posterior framework to the supervised setting, where each observation is a feature–response pair $z = (x, y)$. In this setting, a predictive rule generates future pairs $Z_{n+1}, \ldots, Z_N$. Fong et al. (2023) describe this as the "joint method," which requires modelling both the distribution of $X$ and the conditional distribution $Y \mid X$. Because modelling $X$ can be challenging even in moderate dimensions, we follow Fong et al. (2023) and use the Bayesian bootstrap (see Section 4) to forward sample features, drawing $X_{i+1}$ at random from the empirical distribution of $x_{1:i}$. In our work, we adopt this strategy for $X$, while TabPFN supplies the conditional $Y \mid X$.

### 3.1. Proposed Methodology

We propose instantiating the martingale posterior with the predictive rule induced by TabPFN through in-context learning. TabPFN is a *prior-data-fitted network* (Müller et al., 2022), explicitly trained to approximate a Bayesian PPD by sampling training data from exchangeable sequences defined by likelihood–prior pairs; see Appendix A for a review. We term the resulting martingale posterior **TabMGP**.

Several distinctive features of TabPFN facilitate its integration as a predictive rule within the martingale posterior framework. Through in-context learning, TabPFN consumes the context alongside a new query to return a one-step-ahead predictive distribution without the need for retraining or fine-tuning. This allows the model to adapt to novel datasets instantaneously, handling classification via deterministic probability vectors and regression through a multiclass classification approach that bins the response variable.

The architecture further benefits from native row-permutation invariance, ensuring that predictions depend solely on the multiset rather than the sequence order. This built-in property provides a significant computational advantage over many existing martingale posteriors, such as copula-based constructions (Fong et al., 2023), which must manually enforce invariance by averaging over permutations. Finally, as a prior-data-fitted network, TabPFN is designed to be approximately Bayesian, having been trained to approximate a Bayesian PPD. In the martingale posterior framework, the Bayesian PPD serves as the ideal predictive rule; when the two coincide, the martingale posterior recovers the classical Bayesian posterior. By training on synthetic

---

**Algorithm 1** TabMGP

---

**Require:** Observed data $z_{1:n}$, loss function $\ell(z, \theta)$, forward sampling depth $N$, predictive rule $\text{TabPFN}(\cdot \mid x_{i+1}, z_{1:i})$, number of posterior samples $L$

1: **for** $l = 1$ to $L$ **do**
2:     Set $z_{1:n}^{(l)} \leftarrow z_{1:n}$.
3:     **for** $i = n$ to $N - 1$ **do**
4:         Sample $x_{i+1}^{(l)} \leftarrow \text{Empirical}(x_{1:i}^{(l)})$.
5:         Sample $y_{i+1}^{(l)} \leftarrow \text{TabPFN}(\cdot \mid x_{i+1}^{(l)}, z_{1:i}^{(l)})$.
6:         Form $z_{i+1}^{(l)} \leftarrow (x_{i+1}^{(l)}, y_{i+1}^{(l)})$.
7:     **end for**
8:     Compute $\theta^{(l)} \leftarrow \arg\min_\theta \sum_{i=1}^{N} \ell(z_i^{(l)}, \theta)$.
9: **end for**
10: Return $\{\theta^{(1)}, \ldots, \theta^{(L)}\}$ as samples from the martingale posterior.

---

data from a vast array of likelihood–prior pairs, TabPFN is encouraged to approximate this ideal behaviour across a highly diverse range of tasks.

**Why not other transformers?** Large language models and other pretrained transformers often exhibit in-context learning: given a context $z_{1:i}$ and query $x_{i+1}$, they can produce a prediction for $y_{i+1}$. In this sense, they share the same basic mechanism as TabPFN. However, they lack the two further properties that make TabPFN especially well-suited as a predictive rule for martingale posteriors.

First, they are not row-permutation invariant. Since such models rely on positional embeddings, their predictions change with the ordering of training examples, unless explicit averaging is performed across permutations. Second, they are not trained to approximate a Bayesian PPD: their training objectives are tailored to next-token text prediction rather than calibrated probabilistic prediction for supervised tabular tasks. These differences mean that, although generic transformers can perform in-context learning, their predictive distributions are not naturally aligned with the martingale posterior framework. By contrast, TabPFN combines in-context learning with built-in permutation invariance and Bayesian alignment, making it a more principled choice.

### 3.2. Validity of TabMGP

A central question for any MGP construction is whether the chosen predictive rule leads to a well-defined limiting law $F_\infty$. While the framework traditionally points to conditions like the martingale property (Fong et al., 2023) or the *almost conditionally identically distributed* (a.c.i.d.) condition (Battiston & Cappello, 2025) as sufficient guarantees, verifying these conditions for high-capacity models like TabPFN is currently impossible with existing tools in deep learning theory. Preliminary visual inspections of the predictive rule

associated with TabPFN suggest there may be subtle deviations from these sufficient conditions (see Appendix C). However, we argue that such properties are sufficient rather than necessary for delivering useful epistemic uncertainty.

In the absence of formal theoretical guarantees, we rely on empirical diagnostics to assess the behaviour of TabMGP:

- **Path Stability (Scaled $L_1$ Convergence):** The stability of the functional $\theta(F_N)$ as the rollout length $N$ increases is the primary diagnostic for the existence of a stable limit. We monitor this convergence by tracking the expected $L_1$-norm between the initial empirical risk minimiser $\theta(F_n)$ and the terminal estimate $\theta(F_N)$, scaled by the parameter dimension $p$: $\mathbb{E}_{F_N}[\frac{1}{p}\|\theta(F_n) - \theta(F_N)\|_1]$. If the transformer is internalising a consistent law, these trajectories should stabilise at a non-zero constant as $N$ grows. Systematic drift or divergent $L_1$ paths would indicate that $\theta(F_\infty)$ is ill-defined and the resulting finite-$N$ posterior should be treated cautiously. In practice, we also use this to determine the minimum $N$ needed for convergence.
- **Frequentist Coverage Assessment:** We use frequentist coverage as an empirical check on the relationship between the transformer's predictive rule and the population $F^\star$. A $(1 - \alpha)$ credible set should contain the population risk minimiser $\theta(F^\star)$ at a rate of at least $1 - \alpha$.
- **Posterior Contraction:** A valid epistemic uncertainty measure must represent uncertainty that vanishes as the amount of data increases. One can verify this behaviour by observing the width of the posterior of $\theta(F_N)$ as the observed sample size $n$ increases. A sensible predictive rule should result in a posterior narrowing towards the population risk minimiser $\theta(F^\star)$ as evidence accumulates, reflecting a reduction in epistemic uncertainty (Figure 2).

By focusing on these observed behaviours, practitioners can assess the outputs of TabMGP based on its empirical consistency, bypassing the currently insurmountable challenge of proving the transformer's limiting properties.

**Implications of non-convergence of $F_N$.** If $F_N$ fails to converge, $\theta(F_N)$ may oscillate or diverge. This implies that the martingale posterior, which is defined as the distribution of $\theta(F_\infty)$, is ill-defined. Even when $\theta(F_\infty)$ does exist, practical implementation requires stopping at a finite $N$, meaning the quality of the approximation hinges on the proximity of $\theta(F_N)$ to $\theta(F_\infty)$. Our path stability diagnostic (Appendix F) is designed to detect this. If the $L_1$ trajectory has not plateaued, practitioners should increase $N$.

## 4. Related Work

The martingale posterior framework admits many possible predictive rules, and a growing literature has explored different constructions. We review these constructions below,

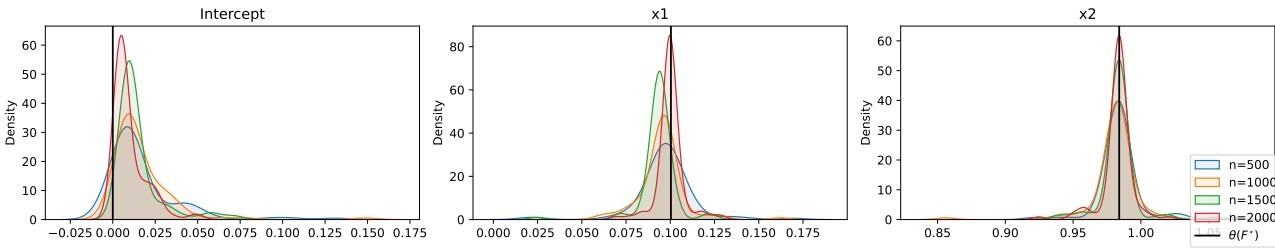

*Figure 2.* Concentration of TabMGP. The black vertical line indicates the population risk minimiser $\theta(F^\star) = (0, \beta^{\star\top})$. See Appendix C for experimental details.

since they form the competing methods for TabMGP in Section 5. In parallel, a separate line of work has asked whether the predictive behaviour induced by transformer in-context learning satisfies martingale properties. This is especially relevant here as TabPFN is a transformer foundation model whose in-context learning defines our predictive rule.

Recent analyses (Falck et al., 2024; Ye & Namkoong, 2024; Nagler & Rügamer, 2025) show that the predictive rules induced by foundation models, such as TabPFN, need not satisfy the classical martingale property. Building on this line of work, we take the further step of assessing the more flexible almost conditionally identically distributed (a.c.i.d.) condition (Battiston & Cappello, 2025) and observe that TabPFN departs from this requirement as well. Nevertheless, we find that the resulting TabMGP exhibits consistent path stability and near-nominal coverage in practice. This suggests that the usual sufficient conditions are stricter than necessary and that validity can be achieved under weaker assumptions than previously theorised.

Employing TabPFN as the predictive rule in the martingale posterior framework has also been considered in Nagler & Rügamer (2025) for conditional inference (whereas we focus on unconditional inference for loss-defined functionals). However, their finding that TabPFN is not a martingale leads them to abandon it as a predictive rule after the initial step, instead using its output to initialise a copula-based update. This approach preserves martingale validity but also inherits the drawbacks of copula-based updates (e.g., fragile bandwidth tuning). By contrast, we retain TabPFN as the predictive rule throughout and interpret its departure from both the martingale and a.c.i.d. conditions as evidence that validity can be achieved under weaker assumptions.

A concurrent effort by Fortini et al. (2026) focuses on posterior inference for $F_\infty \mid z_{1:n}$ using a predictive central limit theorem (CLT). However, their framework does not incorporate the generalised Bayes flavour of inference for a parameter $\theta$ defined by a user-specified loss function. The predictive CLT has distinct advantages in supervised settings, and extending the predictive CLT to functionals $\theta(F_\infty)$ would be an interesting direction for future work.

We now give a brief summary of the main predictive rules developed in the martingale posterior literature.

**Bayesian bootstrap.** The *Bayesian bootstrap* (Rubin, 1981) is a predictive rule where future observations $Z_{i+1}$ are uniformly drawn from the observations up to $i$:

$$Z_{i+1} \sim F_i, \quad i \geq n,$$

with $F_i$ the empirical distribution of $z_{1:i}$. As this rule is discrete, its use is recommended only when the functional of interest arises from a parametric likelihood loss, for which the smoothness of $F_\infty$ is irrelevant (Fong et al., 2023; Dellaporta et al., 2022).

**Newton's algorithm.** Rooted in Bayesian nonparametrics, *Newton's algorithm* (Newton et al., 1998) was initially developed for unsupervised sequential learning in mixture models. It was subsequently re-interpreted as a predictive rule that generates a c.i.d. sequence of observations (Fortini & Petrone, 2020). This approach, however, is not suitable for high-dimensional $\theta$, as it requires numerical integration of an intractable normalising constant.

**Copula-based updates.** For applications requiring smoothness of $F_\infty$, the mainstream implementation of the martingale posterior is dominated by *copula-based updates*. Introduced by Hahn et al. (2018) and adopted as the default predictive rule in Fong et al. (2023), these updates smooth $F_N$ by attaching a bivariate copula to each new observation while preserving the martingale property of the predictive rule exactly. Subsequent work has extended this framework with autoregressive Gaussian-process likelihoods for higher-dimensional features (Ghalebikesabi et al., 2023), direct quantile updates (Fong & Yiu, 2025), log-concave shape constraints (Cui & Walker, 2026), and vine copulas for very high-dimensional features (Huk et al., 2024). All these variants preserve the exact martingale guarantee of the copula framework, but each introduces at least one smoothing or dependence hyperparameter, whose value controls the spread of the resulting posterior. These hyperparameters are generally difficult to tune and must be retuned for each new dataset.

**Parametric plug-in predictives.** For parametric inference, Walker (2022) and Holmes & Walker (2023) proposed a predictive rule of the form $Z_{i+1} \sim p(\cdot \mid \widehat{\theta}_i)$, where $p(\cdot \mid \theta)$ is an assumed parametric model and $\widehat{\theta}_i$ is a point estimate (e.g., maximum likelihood estimate) computed from all previous observations $z_{1:i}$. The asymptotic properties of this predictive rule are studied in Fong & Yiu (2026). However, it requires that the assumed parametric model provides a reasonable approximation to the data distribution in order to yield sensible uncertainty quantification.

**Permutation-equivariant neural network.** Martingale posteriors have also been explored in the context of neural processes (Lee et al., 2023), where a permutation-equivariant neural network is used to construct a martingale posterior over latent variables. While this approach demonstrates the potential of using neural networks within a martingale posterior framework, the focus is still on prediction rather than inference, and it remains unclear how to generalise to broader statistical inference tasks.

## 5. Experiments

We compare TabMGP with other martingale posterior constructions using synthetic and real-world data. Our experiments focus on small- to moderate-$n$ settings where the prior specification of the classical Bayesian posterior is important, making the MGP particularly well suited.

To evaluate a given posterior $\Pi(\cdot \mid z_{1:n})$, we compute the *coverage* and "size" of its joint $(1 - \alpha)$ credible set $C_{1-\alpha}(z_{1:n}) \subset \Theta$:

$$\Pr_{\theta \sim \Pi(\cdot \mid z_{1:n})} \big( \theta \in C_{1-\alpha}(z_{1:n}) \big) = 1 - \alpha.$$

Although the set can be constructed in different ways, an ellipsoidal approximation is often used for computational efficiency; see Appendix D for a discussion.

The coverage of $C_{1-\alpha}(z_{1:n})$ is defined as

$$\Pr_{z_{1:n} \sim F^\star} \big( \theta(F^\star) \in C_{1-\alpha}(z_{1:n}) \big), \tag{4}$$

namely, the probability with which the credible set contains the population risk minimiser, $\theta(F^\star) = \arg\min_\theta \int \ell(z, \theta) \, \mathrm{d}F^\star(z)$, under repeated draws of $z_{1:n}$ from the true data-generating distribution $F^\star$. Although frequentist in nature, this metric is commonly used for comparing Bayesian posteriors.

**Posterior constructions.** We compare TabMGP against the martingale posteriors suggested in Fong et al. (2023): Bayesian bootstrap (*BB*) and the bivariate copula update (*Copula*). We also include two non-martingale baselines built from the Gaussian centred at the loss minimiser $\widehat{\theta}_n$ and

scaled by the inverse-Hessian of the loss: *Asymptotic* uses this Gaussian directly as the credible set, while *Bayes* uses it as the prior in a posterior under a parametric likelihood matched to the loss. The model choice is discussed later in this section when introducing the loss functions. The exact computation details of the posteriors are provided in Appendix E.1. Additional coverage experiments using alternative posterior constructions are reported in Appendix H. These include a standard Bayesian posterior with a diffuse $\mathcal{N}(0, 10^2)$ prior and a TabPFN-initialised copula baseline adapted from Nagler & Rügamer (2025).

**Loss functions.** We focus exclusively on interpretable linear models. We use negative log-likelihood as the loss in (1) and use the corresponding model for the Bayes posterior. For continuous responses (linear regression), we use $\ell(x, y, \theta) = (y - [1 \ x^\top]\theta)^2$, which is the negative log-likelihood of $\mathcal{N}([1 \ x^\top]\theta, 1)$. For $K$-class categorical responses (logistic regression), we use $\ell(x, y, \theta) = -\log \Pr(y = k)$, where $\Pr(y = k)$ is the $k$-th entry of the softmax applied to the logits $([1 \ x^\top]\theta_1, \dots, [1 \ x^\top]\theta_K)$, and $\theta_1, \dots, \theta_K$ are the class-specific coefficients. We also set appropriate constraints on the coefficients to ensure identifiability in the loss function; see Appendix E.2 for details.

**Data-generating setups.** We describe the true data-generating distribution $F^\star$ used here. For each $F^\star$, we draw 100 independent datasets $\{z_{1:n}^{(r)}\}_{r=1}^{100}$ from $F^\star$. We refer to $F^\star$ together with the procedure used to generate $z_{1:n}$ as a *setup*. The sample size $n$ is chosen to be as small as possible without rendering the design matrix ill-conditioned. In total, we have 30 setups (11 synthetic, 19 real-data).

Let $\beta^\star$ be a vector of regression coefficients and let $x_i$ be sampled i.i.d. from the hypercube $[-1, 1]^{10}$. For the synthetic setups, we generate $y \mid x$ with:

1. $y_i = x_i^\top \beta^\star + \epsilon_i$, where $\epsilon_i \overset{iid}{\sim} \mathcal{N}(0, 1)$;
2. $y_i = x_i^\top \beta^\star + \epsilon_i$, where $\epsilon_i$ are i.i.d. draws from Student-$t$ distributions with 5, 4, or 3 degrees of freedom;
3. $y_i = x_i^\top \beta^\star + \epsilon_i$, where $\epsilon_i$ has heteroscedastic variance that depends on $x_i$. We use three strengths of heteroscedasticity: $s_1, s_2, s_3$, from weak to strong;
4. $y_i \mid x_i \overset{iid}{\sim} \mathrm{Bernoulli}(L(x_i^\top \beta^\star))$, with a logistic link function $L$;
5. $y_i \mid x_i \overset{iid}{\sim} \mathrm{Bernoulli}(L(x_i^\top \beta^\star))$, where $L$ is the distribution function of a Gaussian mixture model (GMM). We consider three choices of location parameters: GMM(0), GMM(−1), and GMM(−2), where the argument denotes the mean of one mixture component.

For the 19 real-data setups, we use datasets from the OpenML and UCI repositories (9 continuous response, 8 binary response, and 2 multiclass categorical response). As $F^\star$ is unknown in the real-data setups, we take the empirical

distribution of the entire dataset as the truth $\widetilde{F}^\star$. The details of all data-generating setups are given in Appendix E.3. We also include experiments that combine real covariates with known nonlinear response mechanisms, providing a complementary check between the fully synthetic and real-data regimes; see Appendix G.

**Metrics for comparing posteriors.** For each posterior, we construct joint credible sets $\widehat{C}_{1-\alpha}(z_{1:n})$ and compare their coverage (4) and "size". Our approach to constructing $\widehat{C}_{1-\alpha}(z_{1:n})$ is described in Appendix E.4. We consider a posterior more reliable and meaningful when its credible set is small while achieving coverage at least $1-\alpha$.

In the synthetic setups, the coverage is given by $\frac{1}{100}\sum_{r=1}^{100}\mathbf{1}[\theta(F^\star)\in\widehat{C}_{1-\alpha}(z_{1:n}^{(r)})]$, where $\{z_{1:n}^{(r)}\}_{r=1}^{100}$ are 100 replicates of $z_{1:n}$ drawn from $F^\star$. In the real-data setups, as $F^\star$ is unknown, we take the empirical distribution of the entire dataset $\mathcal{D}$ as the truth $\widetilde{F}^\star$, and compute the coverage on $\theta(\widetilde{F}^\star)=\arg\min_\theta\sum_{z\in\mathcal{D}}\ell(z,\theta)$. That is, the coverage is $\frac{1}{100}\sum_{r=1}^{100}\mathbf{1}[\theta(\widetilde{F}^\star)\in\widehat{C}_{1-\alpha}(z_{1:n}^{(r)})]$, where $z_{1:n}^{(r)}$ denotes the $r$-th random draw (without replacement) from $\mathcal{D}$; see Appendix E.3 for details.

We use the trace of the posterior covariance matrix as a proxy for the "size" of a credible set.

### 5.1. Results

**Convergence of $\theta(F_N)$.** To conduct statistical inference, we require $\theta(F_N)$ to converge $\mathbb{P}$-a.s. as $N\to\infty$. In practice, for a fixed observed dataset $z_{1:n}$, we approximate this limit by drawing $T$ additional forward-sampling steps, so that the rollout length is $N=n+T$. This convergence is commonly assessed using trace plots of each component of $\theta(F_N)$, but this approach quickly becomes impractical in high dimensions. Instead, we track the expected $L_1$-norm of $\theta(F_N)$ relative to the empirical risk minimiser $\theta(F_n)$ as $T$ increases; see Appendix F for the precise definition. For a convergent $\theta(F_N)$, this diagnostic should stabilise at a non-zero constant as $T$ increases.

We show these trajectories in Figure 3, where each trajectory corresponds to one of the 30 setups. For completeness, we also report the $L_1$-norm between each realisation of $\theta(F_N)$ and $\theta(F_n)$ in Figure 6. In both Figure 3 and 6, the diagnostic is computed from a single realisation of $z_{1:n}$, reflecting the practical setting in which an analyst has access to only one observed dataset. We also report the same diagnostic over 20 independent realisations of $z_{1:n}$ for each setup in Figure 8.

In general, we observe that $T=500$ forward-sampling steps is sufficient for convergence in most setups. For slower-converging setups, we use longer forward-sampling runs and verify convergence by $T=1000$; see Figure 7. Unless stated otherwise, all subsequent results use $T=500$.

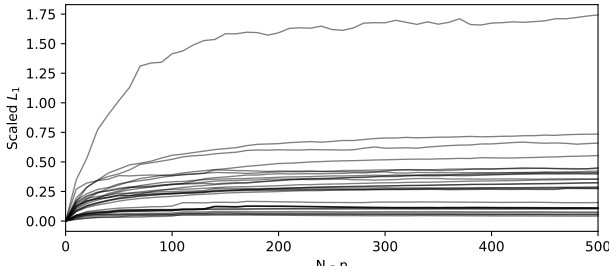

*Figure 3.* Expected $L_1$-norm between $\theta(F_N)$ from TabMGP and $\theta(F_n)$ as $N$ increases. Each of the 30 trajectories corresponds to a realisation of $z_{1:n}$ from one setup.

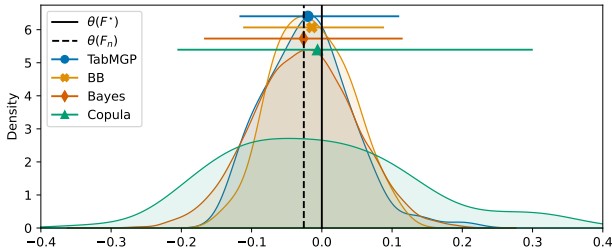

*Figure 4.* Posterior densities of the intercept in the 'concrete' setup. For each density, the 95% marginal credible interval is shown as a horizontal bar, and the posterior mean is marked. The solid and dashed vertical lines correspond to $\theta(F^\star)$ and $\theta(F_n)$, respectively.

**Shape of the posteriors.** We present the marginal densities of the posteriors for a realisation of $z_{1:n}$ in Figure 4 (intercept of the 'concrete' setup), with the extra dimensions and setups provided in Appendix J. In general, the posterior means are similar across methods, but TabMGP often exhibits skewness and multimodal structure compared with the other posteriors, which are typically Gaussian-like.

**Joint credible sets for linear regression.** The results for synthetic data (top and middle) and real-data (bottom) setups are shown in Table 1. In the real-data setups, the rows are ordered by decreasing $n/p$, with $p := \dim(\theta)$.

TabMGP is competitive in both coverage and credible set size, performing particularly well on real data. Notably, it can attain nearly 100% coverage in many setups while maintaining small credible sets. Similarly, Copula provides good coverage with small credible sets, although its performance is less consistent across real-data setups (e.g., severe undercoverage in kin8nm and overcoverage in quake). This is unsurprising, as bespoke Gaussian-copula-based predictive rules have an advantage in Gaussian (or near-Gaussian) setups but can be sensitive to departures from that structure. Bayes and Asymptotic produce excessively large credible sets in all synthetic setups, as both rely on asymptotic approximations that are less accurate in low-$n$ settings. They are also more susceptible to model misspecification and

*Table 1.* Experiments on linear regression: i.i.d. errors (top), dependent errors (middle), and real data (bottom). We compare TabMGP (ours), BB, Copula, Bayes and Asymptotic. The rows corresponding to real data are sorted by $n/p$ ratio, i.e., bottom rows are harder. Reported values are coverage (Rate) and the median trace of the posterior covariance (Size), computed over 100 repetitions. Target coverage is 0.95.

| | TabMGP | | BB | | Copula | | Bayes | | Asymptotic | |
|---|---|---|---|---|---|---|---|---|---|---|
| Setup | Rate | Size | Rate | Size | Rate | Size | Rate | Size | Rate | Size |
| $\mathcal{N}(0,1)$ | 1.00 | 0.45 | 0.55 | 0.09 | **0.99** | **0.35** | 1.00 | 0.65 | 1.00 | 1.31 |
| $t_5$ | 1.00 | 0.49 | 0.65 | 0.11 | **0.99** | **0.37** | 1.00 | 0.65 | 1.00 | 1.31 |
| $t_4$ | 0.99 | 0.51 | 0.64 | 0.12 | **0.98** | **0.37** | 0.99 | 0.65 | 1.00 | 1.31 |
| $t_3$ | 1.00 | 0.48 | 0.66 | 0.14 | **0.97** | **0.35** | 0.98 | 0.65 | 0.98 | 1.31 |
| $s_1$ | 1.00 | 0.37 | 0.50 | 0.03 | **1.00** | **0.36** | 1.00 | 0.65 | 1.00 | 1.31 |
| $s_2$ | **1.00** | **0.36** | 0.46 | 0.02 | 1.00 | 0.36 | 1.00 | 0.65 | 1.00 | 1.31 |
| $s_3$ | **1.00** | **0.33** | 0.53 | 0.02 | 1.00 | 0.37 | 1.00 | 0.65 | 1.00 | 1.31 |
| concrete | 0.91 | 0.06 | 0.80 | 0.05 | 1.00 | 0.12 | 0.87 | 0.05 | **1.00** | **0.10** |
| quake | 1.00 | **0.03** | 0.95 | 0.07 | 1.00 | 0.18 | 0.91 | 0.04 | 0.98 | 0.09 |
| airfoil | 0.96 | 0.08 | 0.93 | 0.05 | 0.97 | 0.11 | **0.96** | **0.06** | 1.00 | 0.12 |
| energy | **1.00** | **0.04** | 0.80 | 0.01 | 1.00 | 0.06 | 1.00 | 0.07 | 1.00 | 0.14 |
| fish | 0.90 | 0.10 | 0.79 | 0.07 | **0.95** | **0.10** | 0.91 | 0.08 | 1.00 | 0.16 |
| kin8nm | **0.95** | **0.13** | 0.77 | 0.09 | 0.63 | 0.08 | 0.84 | 0.11 | 0.99 | 0.22 |
| auction | **0.99** | **0.66** | 0.86 | 0.35 | 0.97 | 1.44 | 1.00 | 0.81 | 1.00 | 1.65 |
| grid | **0.97** | **0.13** | 0.72 | 0.08 | 0.88 | 0.10 | 0.97 | 0.16 | 1.00 | 0.31 |
| abalone | **0.97** | **0.95** | 0.66 | 0.69 | 0.94 | 0.98 | 0.84 | 0.81 | 0.98 | 1.64 |

*Table 2.* Experiments on logistic regression: synthetic data with various link functions (top), real data with binary logistic regression (middle), and real data with multinomial logistic regression (bottom). The rows corresponding to real data are sorted by $n/p$ ratio, i.e., bottom rows are harder. We compare TabMGP (ours), BB, Copula, Bayes and Asymptotic. Reported values are coverage (Rate) and the median trace of the posterior covariance (Size), computed over 100 repetitions. Target coverage is 0.95.

| | TabMGP | | BB | | Copula | | Bayes | | Asymptotic | |
|---|---|---|---|---|---|---|---|---|---|---|
| Setup | Rate | Size | Rate | Size | Rate | Size | Rate | Size | Rate | Size |
| Logistic | 0.84 | 1.30 | 0.83 | 1.96 | 1.00 | 2.72 | 0.53 | 0.78 | **0.96** | **1.53** |
| GMM(0) | 0.91 | 2.44 | 0.91 | 4.14 | 1.00 | 4.01 | 0.60 | 1.34 | **0.98** | **2.63** |
| GMM(-1) | 0.83 | 1.27 | 0.86 | 1.95 | 1.00 | 2.64 | 0.58 | 0.79 | **0.99** | **1.54** |
| GMM(-2) | 0.78 | 0.88 | 0.78 | 1.36 | 1.00 | 2.08 | 0.52 | 0.57 | **1.00** | **1.13** |
| rice | **1.00** | **3.74** | 0.88 | 7.35 | 1.00 | 28.41 | 0.91 | 2.87 | 1.00 | 5.92 |
| sepsis | **0.95** | **0.69** | 1.00 | 2.42 | 1.00 | 10.00 | 0.98 | 1.18 | 1.00 | 2.42 |
| banknote | 0.99 | 1.16 | 1.00 | 1.48 | 1.00 | 1.53 | **0.99** | **0.64** | 1.00 | 1.28 |
| mozilla | **0.96** | **1.42** | 0.97 | 2.04 | 1.00 | 2.68 | 0.87 | 0.65 | 0.93 | 1.31 |
| skin | 1.00 | 1.39 | 1.00 | 1.84 | 1.00 | 1.80 | **1.00** | **0.57** | 1.00 | 1.16 |
| blood | 0.87 | 1.07 | 0.95 | 2.51 | 1.00 | 5.40 | 0.84 | 0.86 | **0.99** | **1.71** |
| phoneme | **1.00** | **1.32** | 0.95 | 1.92 | 1.00 | 5.62 | 0.73 | 0.83 | 1.00 | 1.66 |
| telescope | **0.99** | **5.96** | 0.99 | 10.66 | 1.00 | 50.69 | 0.67 | 3.50 | 0.96 | 7.10 |
| yeast | **0.97** | **11.10** | 0.68 | 25.13 | – | – | 0.33 | 10.82 | 0.88 | 19.22 |
| wine | **0.93** | **11.04** | 0.79 | 39.75 | – | – | 0.09 | 12.51 | 0.91 | 26.25 |

tend to underperform relative to TabMGP. In contrast, BB tends to produce overly small sets with poor coverage in the synthetic setups (top and middle) and in low $n/p$ settings (last few rows of Table 1). This reflects limited variability in forward samples when bootstrapping from a small $z_{1:n}$ (e.g., $n = 20$ in the synthetic data setups).

**Joint credible sets for logistic regression.** The results for synthetic (top) and real-data (middle and bottom) setups are shown in Table 2. Copula was not applied to multinomial logistic regression (bottom) as the implementation of Fong et al. (2023) is not applicable. In the real-data setups, the rows are ordered by decreasing $n/p$ within each section.

In the synthetic setups, the Asymptotic credible sets achieve the best coverage and size, as expected, since the asymptotic approximation is reasonably accurate at the larger sample size ($n = 100$). TabMGP, however, shows the most consistent performance in the real-data setups, often achieving good coverage with the smallest credible sets. TabMGP performs particularly well relative to competing methods in the 'harder' setups, i.e., the bottom rows with low $n/p$ ratios. Asymptotic also provides reasonable coverage in the real-data setups, although its credible sets are typically much wider than those of TabMGP. This is likely due to model misspecification, which results in miscalibrated credible sets. BB performs reasonably well in binary logistic regression (top and middle), but its credible sets are larger than those of TabMGP. However, BB severely undercovers in multinomial logistic regression (bottom), where $p$ is much larger than for the other logistic regression setups. Copula performs poorly in logistic regression, producing overly

large credible sets. This suggests that Copula is sensitive to the choice of bandwidth, as the same value yields good performance in the linear regression experiment.

**Sensitivity to the rollout length.** TabMGP inherits a single tuning parameter from the martingale posterior framework: the rollout length $N$. Ideally, $N$ should be large enough that the resulting credible sets are insensitive to additional forward sampling, although computational constraints can limit this in practice. Appendix I assesses the sensitivity of these credible sets by varying $N$ in the slower-converging setups. The results show that both coverage and credible-set size stabilise once $N$ is sufficiently large.

**Marginal credible intervals.** In addition to the results of joint credible sets presented above, we also report marginal credible-interval results in Appendix K. These one-dimensional intervals provide an alternative assessment of the coverage that does not rely on the empirical-cutoff ellipsoids used for the joint credible sets. We report both the coverage and the median width of the credible intervals. The results are broadly consistent with the results from the joint credible sets above.

**Computation time.** The main computational cost of martingale posteriors is forward sampling, which is readily parallelisable. Per sample, BB and Asymptotic are the fastest (near-instantaneous); Copula takes 10–20s; and TabMGP takes about 70s (binary/categorical response) or 200s (continuous response). However, despite being the most computationally expensive method, TabMGP requires minimal manual intervention compared to the others. The No-U-

Turn algorithm (Hoffman & Gelman, 2014) used for Bayes runs in under 60s. BB, Copula, Bayes, and Asymptotic are implemented in compiled JAX, whereas TabMGP currently runs in uncompiled PyTorch. We expect this gap to narrow with further code optimisation and improvements in the inference speed of tabular foundation models. All experiments were conducted on an NVIDIA L40S.

## 6. Conclusion

While bespoke Bayesian modelling remains the gold standard when a well-specified likelihood–prior pair is available, tabular foundation models like TabPFN are increasingly used in practice. In scientific workflows, prediction is usually a means to an end: one ultimately wants uncertainty about a scientific estimand $\theta$. However, TabPFN does not by itself return posterior uncertainty for $\theta$. In TabPFN, the carrier of epistemic uncertainty is the *predictive rule itself* and the associated random limiting measure $F_\infty$, so the natural inference target is the functional $\theta(F_\infty)$. This motivates a framework that returns posterior uncertainty for $\theta(F_\infty)$ using only queries to TabPFN's predictive outputs.

For this purpose, we introduce TabMGP. The procedure requires neither likelihood–prior specification, as in classical Bayesian inference, nor bespoke predictive rules or hyperparameter tuning, as in existing martingale posterior frameworks, relying instead solely on the predictive rule induced by TabPFN. We evaluated TabMGP across a large collection of synthetic and real-world setups and compared it extensively with existing martingale posterior constructions as well as standard Bayesian baselines. Across these experiments, TabMGP consistently performs well from a frequentist perspective and delivers stable posteriors with near-nominal coverage. Our empirical evaluation shows that TabMGP provides reliable uncertainty quantification.

The experiments also reveal concrete failure modes. TabMGP can exhibit undercoverage in classification problems with small $n/p$ and can exhibit slow convergence of $\theta(F_N)$ in some setups. Accordingly, practitioners should treat TabMGP as a diagnostic-driven procedure. Under finite computational budgets, one should inspect the path-stability diagnostics described in Appendix F and interpret the posterior with non-stabilising paths cautiously.

Our findings also expose limitations in the current theoretical understanding of martingale posteriors. Existing analyses emphasise sufficient conditions such as the martingale or a.c.i.d. properties; in our experiments, finite-horizon diagnostics do not *certify* these sufficient conditions, yet the resulting posteriors maintain empirical stability and near-nominal coverage. This suggests that the present theory may be overly restrictive and does not yet capture the range of predictive rules that work well empirically. Developing

weaker and more realistic conditions is an important direction for future research, particularly as martingale posteriors are likely to be combined with an increasing variety of black-box predictive models. More broadly, deep learning theory continues to lag far behind practice. For predictors like TabPFN, we should not expect to be able to verify theoretically that any proposed condition holds, so progress will also require a battery of theory-guided empirical diagnostics to test these conditions.

## Acknowledgements

The authors would like to thank the reviewers and area chair for their helpful comments and suggestions. We also thank Abdelhamid Ezzerg, Joshua Bon, Christian Robert, Lorenzo Cappello, Jack Jewson, Rubén Loaiza-Maya and Anastasios Panagiotelis for helpful comments in the development of this work. KN was supported by the Australian Government Research Training Program and the Statistical Society of Australia PhD Top-up Scholarship. EF was supported by the Research Grants Council of Hong Kong through the General Research Fund (17307321) and the Early Career Scheme (27304424). DTF acknowledges funding from the Australian Research Council (DE200101070, DP200101414). JK was supported through the UK's Engineering and Physical Sciences Research Council (EPSRC) via EP/W005859/1. SW was supported by the Australian Research Council (DE200101253). Computational resources were provided by the ARDC Nectar Research Cloud.

## Impact Statement

This paper presents work whose goal is to advance the field of Machine Learning. There are many potential societal consequences of our work, none of which we feel must be specifically highlighted here.

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

## A. Review of TabPFN

A precursor to TabPFN appears in Müller et al. (2022), where the transformer is referred to as a prior-data-fitted network (PFN). The underlying premise is that the Bayesian posterior predictive distribution (PPD) is optimal in an *average* sense. We present this result in the unsupervised setting to keep the notation lightweight, drawing on a result from Aitchison (1975)[1].

Suppose we have an exchangeable distribution

$$p(z_1, \ldots, z_n) = \int \prod_{i=1}^{n} p(z_i \mid \nu)\phi(\nu) \, \mathrm{d}\nu \,. \tag{5}$$

Consider the Kullback-Leibler (KL) divergence between $p(z \mid \nu)$ and some predictive density $q(z \mid z_{1:n})$ where $z_{1:n}$ is drawn from (5), i.e., first draw $\nu$ from the prior $\phi(\nu)$ and then draw $z_i$ i.i.d. from $p(z \mid \nu)$. Then,

$$D_{\mathrm{KL}}(p(z \mid \nu) \,\|\, q(z \mid z_{1:n}))$$

is random because (a) $\nu$ is random and (b) the data $z_{1:n}$ are random. By integrating out these sources of randomness, we get an average KL divergence, a deterministic quantity

$$\mathbb{E}_{\nu, z_{1:n}} D_{\mathrm{KL}}(p(z \mid \nu) \,\|\, q(z \mid z_{1:n})) = \int \phi(\nu) \, \mathrm{d}\nu \int p(z_{1:n} \mid \nu) \, \mathrm{d}z_{1:n} \int p(z \mid \nu) \log \frac{p(z \mid \nu)}{q(z \mid z_{1:n})} \, \mathrm{d}z \,. \tag{6}$$

Aitchison (1975) then shows that the predictive density $q$ that minimises this average KL divergence in (6) is none other than the Bayesian PPD corresponding to the likelihood–prior pair in (5):

$$p(z \mid z_{1:n}) = \int p(z \mid \nu)\pi(\nu \mid z_{1:n}) \, \mathrm{d}\nu \tag{7}$$

where $\pi(\nu \mid z_{1:n}) \propto \prod_{i=1}^{n} p(z_i \mid \nu)\phi(\nu)$, the standard Bayesian posterior construction.

This optimality result for the Bayesian PPD immediately suggests a meta-learning training objective. Since minimising the average KL divergence with respect to $q$ is indifferent to the entropy term of $p(z \mid \nu)$ in the KL divergence, we may rewrite the objective as

$$\underset{q(z|z_{1:n})}{\arg\min} \, \mathbb{E}_{\nu, z_{1:n}} \, \mathrm{KL}(p(z \mid \nu) \,\|\, q(z \mid z_{1:n})) = \underset{q(z|z_{1:n})}{\arg\min} \, \mathbb{E}_{\nu, z_{1:n}} \mathbb{E}_{p(z|\nu)} - \log q(z \mid z_{1:n})$$

If we use a model $q_w$, the empirical risk counterpart is

$$\underset{w}{\arg\min} - \sum_{m=1}^{M} \sum_{k=1}^{K} \log q_w(z_{n+k}^m \mid z_{1:n}^m).$$

Here, $q_w$ can be any model that can accept a variable-length set $z_{1:n}$ and returns a distribution over future $z$, and the samples $z_{1:n+K}$ are drawn from an exchangeable distribution $p(z_1, \ldots, z_n, z_{n+1}, \ldots, z_{n+K})$ as in (5). Müller et al. (2022) choose to use a transformer for $q_w$. This is most natural when $z$ is categorical. In the case of real-valued $z$, Müller et al. (2022) perform an adaptive binning operation that essentially converts regression into a classification task.

Later developments led to TabPFN (Hollmann et al., 2022; 2025), which is trained on a vast amount of synthetic data generated from a very large array of likelihood–prior pairs (e.g., structural causal models and Bayesian neural network models), rather than a single pair as in the original PFN work. This gives TabPFN its *foundation model* qualifier.

## B. Recovering the Classical Bayesian Posterior from the Martingale Posterior

A sample from the classical Bayesian posterior $\pi(\theta \mid z_{1:n}) \propto \pi(\theta) \prod_{i=1}^{n} p(z_i \mid \theta)$ can be obtained equivalently from the martingale posterior. This is achieved by forward sampling with the Bayesian PPD:

$$p(z \mid z_{1:i}) = \int p(z \mid \theta) \, \pi(\theta \mid z_{1:i}) \, \mathrm{d}\theta,$$

---

[1]The work of Müller et al. (2022) does not appear to be aware of this result.

then computing the posterior mean estimator $\mathbb{E}[\theta \mid z_{1:\infty}]$. This follows from Doob's martingale convergence theorem (Doob, 1949). Alternatively, we can use the maximum likelihood estimator as our functional of interest $\theta(F)$, i.e., by setting the loss in (1) as $\ell(z, \theta) = -\log p(z \mid \theta)$, and the distribution of $\theta(F_\infty)$ is $\pi(\theta \mid z_{1:n})$. This is due to the asymptotic equivalence between a posterior mean estimator and the maximum likelihood estimator.

## C. Validity of TabMGP as a Martingale Posterior

As discussed in Section 2, a sufficient condition for the existence of $F_\infty$ and hence for the martingale posterior distribution to be well-defined is that the sequence $(P_i)_{i\geq 0}$ is a martingale, or equivalently, that the sequence $(Z_i)_{i\geq 1}$ is conditionally identically distributed (c.i.d. Berti et al., 2004). The martingale property stipulates that $(P_i)_{i\geq 0}$ satisfies, for every $i \geq 0$ and every measurable set $A$:

$$\mathbb{E}(P_{i+1}(A) \mid Z_{1:i}) = P_i(A). \tag{8}$$

A relaxation of this sufficient condition is the *almost conditionally identically distributed* (a.c.i.d.) condition (Battiston & Cappello, 2025). Let $(\xi_i)_{i\geq 1}$ be a sequence of non-negative random variables. The sequence $(Z_i)_{i\geq 1}$ is a.c.i.d. if the corresponding predictive rule $(P_i)_{i\geq 0}$ satisfies

$$|\mathbb{E}_{Z_{i+1}}(P_{i+1}(A) \mid Z_{1:i}) - P_i(A)| \leq \xi_i, \quad \text{a.s.,} \tag{9}$$

for all $i \geq 1$ and all measurable sets $A$. Battiston & Cappello (2025) show that $F_\infty$ exists if $\sum_{i=0}^{\infty} \xi_i < \infty$, $\mathbb{P}$-a.s., and that the sequence $(Z_i)_{i\geq 1}$ is $F_\infty$-asymptotically exchangeable. We will refer to these conditions collectively as the *a.c.i.d. condition*. Clearly, (9) generalises the stronger martingale condition (8), i.e., $\xi_i = 0$ for all $i \geq 1$, and we will use (9) as an empirical diagnostic for TabMGP instead of (8).

Verifying (9) empirically is generally challenging, as it must hold for all measurable sets $A$. Instead, we adopt an equivalent formulation in terms of the total variation distance (Battiston & Cappello, 2025):

$$\sum_{i=1}^{\infty} \text{TV}\left(\mathbb{E}_{Z_{i+1}}(P_{i+1}(\cdot) \mid Z_{1:i}), P_i(\cdot)\right) \leq \xi_i, \quad \text{a.s.,}$$

for all $i \geq 0$. Since the predictive distributions of both TabPFN and the Bayesian bootstrap are probability mass functions $p_i$, the condition can be further simplified in terms of the $L_1$ distance:

$$\sum_{i=1}^{\infty} \sum_{z \in \mathcal{Z}} |\mathbb{E}_{Z_{i+1}}[p_{i+1}(z) \mid Z_{1:i}] - p_i(z)| < \infty, \quad \text{a.s..} \tag{10}$$

There are several practical considerations when estimating (10). First, we start the outer summation from $i = n$, as TabPFN requires a small number of initial observations to produce reliable predictions. Second, in our supervised learning setting where $z = (x, y)$, the expectation

$$\mathbb{E}_{Z_{i+1}}[p_{i+1}(z) \mid z_{1:i}] = \mathbb{E}_{Y_{i+1}, X_{i+1}}[p_{i+1}(y \mid x_{i+2}) \, p_{i+1}(x) \mid z_{1:i}],$$

requires a large number of Monte Carlo samples and is computationally demanding. To mitigate this, we fix all future covariates $x_{n+1:N+2}$ to a constant value $x^\star$ and consider the weaker condition:

$$\sum_{i=n}^{N} \sum_{y \in \mathcal{Y}} |\mathbb{E}_{Y_{i+1}}[p_{i+1}(y \mid x^\star) \mid z_{1:i}] - p_i(y \mid x^\star)| < \infty, \tag{11}$$

for a sufficiently large $N$. We then compute (11) repeatedly over many realisations of $Y_{n+1:N}$.

We evaluate this condition on an initial dataset of size $n = 100$ generated as follows: $x_i$ are drawn independently and uniformly from the two-dimensional cube $[-1, 1]^2$, and $y_i \overset{\text{i.i.d.}}{\sim} \text{Bernoulli}(\text{logistic}(x_i^\top \beta^\star))$, for $i = 1, \ldots, n$, with a fixed $\beta^\star \in \mathbb{R}^2$. We consider 10 different values of $x^\star$, randomly selected from the initial sequence $x_1, \ldots, x_n$. Due to computational constraints, we evaluate (11) only on a sparse grid $i \in \{100, 120, 140, \ldots, 1100\}$ and use the trapezoidal rule to approximate the partial sum.

Our empirical results (Figure 5) suggest that the predictive rule induced by TabPFN does not strictly adhere to the a.c.i.d. condition as $N$ increases. However, at these scales ($N > 1000$), any measurable drift falls below the numerical resolution

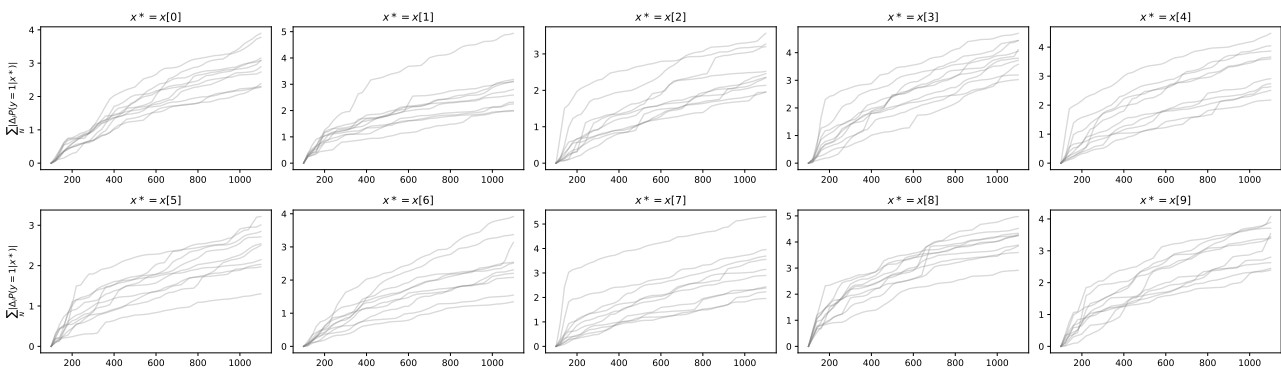

*Figure 5.* Cumulative sum (over $N$) of the $L_1$-distance between $\mathbb{E}_{y_{i+1}}[p_{i+1}(y \mid x^\star) \mid z_{1:i}]$ and $p_i(y \mid x^\star)$ across ten choices of $x^\star$. Ideally, this cumulative sum converges to satisfy the a.c.i.d. condition.

of the model, making it impossible to distinguish genuine theoretical departures from the cumulative numerical instability inherent to long-horizon autoregressive inference. This empirical ambiguity, however, does not imply non-convergence to $F_\infty$ for TabMGP, as the a.c.i.d. condition is sufficient but not necessary. Moreover, the observed divergence may be attributable to the finite numerical precision of TabPFN. In particular, both $\mathbb{E}_{Y_{i+1}}[p_{i+1}(y \mid x^\star) \mid z_{1:i}]$ and $p_i(y \mid x^\star)$ are expected to be accurate only up to machine precision, so that their difference behaves as random numerical noise for sufficiently large $N$, and the partial sum is infeasible to compute in this regime.

As an alternative, we track the sample path of $\theta(F_N)$ as $N$ increases, as in Fong et al. (2023). After all, we need $\theta(F_N)$ to converge $\mathbb{P}$-a.s. to conduct meaningful statistical inference. We refer to these diagnostic plots as the "trace plot" and inspect them in Appendix F. We observe that the sample path of $\theta(F_N)$ does converge in most of our experiments, thus suggesting that the a.c.i.d. condition is too restrictive for our purposes in establishing the existence of $\theta(F_\infty)$.

For completeness, we further demonstrate that TabMGP can concentrate close to $\theta(F^\star)$ as the sample size $n$ grows. To illustrate this, we use a regression dataset, since the sample size needed for meaningful concentration, together with the forward-sampling steps required for convergence in classification, exceeds TabPFN's 10,000-context length limit. The regression dataset is defined as $y_i \overset{i.i.d.}{\sim} \mathcal{N}(x_i^\top \beta^\star, 0.1^2)$, $i = 1, \ldots, n$, for some fixed $\beta^\star \in \mathbb{R}^2$, with the loss function $\ell(x, y, \theta) = (y - [1\ x^\top]\theta)^2$. Ideally, TabMGP should concentrate around $\theta(F^\star) = [0, \beta^{\star\top}]$ as $n$ increases. Figure 2 confirms that TabMGP indeed concentrates around $\theta(F^\star)$.

## D. Highest-Posterior Density Credible Sets

The *highest-posterior-density* (HPD) credible set is a common construction of a Bayesian credible set. The HPD set at level $1 - \alpha$ is defined as

$$C_{1-\alpha}^{\mathrm{HPD}} = \left\{ \theta \in \mathbb{R}^p : \pi(\theta \mid z_{1:n}) \geq t_{1-\alpha} \right\},$$

where $t_{1-\alpha}$ is chosen such that $\int_{C_{1-\alpha}^{\mathrm{HPD}}} \pi(\theta \mid z_{1:n}) \, \mathrm{d}\theta = 1 - \alpha$. By construction, $C_{1-\alpha}^{\mathrm{HPD}}$ has minimum volume among all sets of posterior probability $1 - \alpha$, making it the "gold standard" for summarising joint uncertainty (Hyndman, 1996).

In many models — particularly linear and logistic regressions with well-behaved posteriors — the posterior is approximately multivariate normal:

$$\theta \mid z_{1:n} \approx \mathcal{N}_p(\hat{\mu}, \hat{\Sigma}).$$

In these cases, the HPD set reduces to the $\chi^2$-ellipsoid

$$C_{1-\alpha}^{\mathrm{HPD}} \approx \left\{ \theta : (\theta - \hat{\mu})^\top \hat{\Sigma}^{-1}(\theta - \hat{\mu}) \leq \chi_{p,1-\alpha}^2 \right\}, \tag{12}$$

treating the Mahalanobis ellipsoid as the HPD set under the normal approximation.

## E. Experimental Settings and Hyperparameters

The code is released at `https://github.com/weiyaw/tabmgp`.

### E.1. Posterior Constructions

**Asymptotic approximation of the Bayesian posterior.** Let $\widehat{\theta}_n = \arg\min_\theta \sum_{i=1}^n \ell(z_i, \theta)$ and $\widehat{H}_n = \nabla_\theta^2 \sum_{i=1}^n \ell(z_i, \theta)|_{\theta=\widehat{\theta}_n}$. We use $\mathcal{N}(\widehat{\theta}_n, \widehat{H}_n^{-1})$ as the asymptotic posterior approximation for constructing credible sets and as the prior for the classical Bayes baseline.

**Bayes settings.** We approximate the posterior using a No-U-Turn sampler (NUTS, Hoffman & Gelman, 2014) with 1000 warm-up steps, followed by 2000 sampling steps across 4 chains with different initialisations, for a total of 8000 samples.

**Settings common to all martingale posteriors.** We run 2000 forward-sampling steps for both BB and Copula. For TabMGP, we find that $N = n + 500$ is sufficient for $\theta(F_N)$ to converge (Figure 3 and Appendix F) and use only 500 forward-sampling steps. We approximate all martingale posterior constructions with 100 samples.

**TabMGP settings.** The predictive outputs are obtained from `TabPFNClassifier.predict_proba` for logistic regression and `TabPFNRegressor.predict(output_type = "full")` for linear regression. By default, the corresponding logits are scaled by a temperature of 0.9. For our experiments, we use the distributions induced by the untempered logits as our predictive distributions, i.e., setting the temperature to 1. By construction, TabPFN is not permutation invariant to the ordering of the response and feature dimensions. The default implementation of TabPFN uses an ensemble of predictions obtained by randomly permuting these dimensions. The resulting probability mass functions are then averaged to obtain the final predictive mass function. We use the default number of ensemble members in version `2.0.6`, which is 4 for classifiers and 8 for regressors.

We use the following two checkpoints:

- Classifier: `https://huggingface.co/Prior-Labs/TabPFN-v2-clf/blob/main/tabpfn-v2-classifier.ckpt`

- Regressor: `https://huggingface.co/Prior-Labs/TabPFN-v2-reg/blob/main/tabpfn-v2-regressor.ckpt`

These are the default checkpoints when we ran our experiments, and the transformer was pre-trained exclusively on synthetic, exchangeable sequences of data. Subsequent iterations of the TabPFN checkpoints, however, have been fine-tuned on real-world datasets, and these checkpoints are not used in our experiments.

**Copula settings.** We use the copula construction in Fong et al. (2023, §4.4.2 and §4.4.3) with the bandwidth fixed to 0.8, since for our experiments the marginal-likelihood-based tuning procedure of Fong et al. (2023) produces values that are too small and delivers poor results. Instead, we estimate

$$\theta(P_N) \approx \arg\min_\theta \sum_i \ell(\widetilde{x}_i, \widetilde{y}_i, \theta),$$

where $(\widetilde{x}_i, \widetilde{y}_i)_{i \geq 0}$ are generated by iterating $\widetilde{x}_i$ over $x_{1:n}$ and, for each $\widetilde{x}_i$, drawing $\widetilde{y}_i$ from $P_N(\cdot \mid \widetilde{x}_i)$. We repeat this procedure five times, yielding $5n$ pairs of $(\widetilde{x}_i, \widetilde{y}_i)$.

### E.2. Specification of the Loss Function

We set the following coefficients to 0 to ensure identifiability in the loss function:

1. The coefficients for the first level of each categorical feature;

2. The coefficients corresponding to all (almost) collinear features;

3. In logistic regression, the coefficients corresponding to the first category, $\theta_1$.

We identify the collinear features as those with the largest loading in the least important principal component of the population data, removing them one at a time until the computation of $\theta(F^\star)$ is numerically stable. The resulting parameter dimension, $p := \dim(\theta)$, and the features used in the functional are listed in Table 3 and Table 4, respectively.

We then standardise all continuous responses and features using the population mean and variance, and one-hot encode all categorical features. Using population moments for standardisation ensures that $\theta$ remains on a common scale across repetitions.

### E.3. Data Setups

**Synthetic data.** The 11 synthetic setups are adapted from Wu & Martin (2023, Section 5): the features $x_i$ are i.i.d. and uniformly sampled from the hypercube $[-1,1]^{10}$. The true coefficients $\beta^\star$ are sampled once from $[-2,3]^{10}$ and held fixed across the 100 repetitions of $z_{1:n}$. The response $y \mid x$ is generated as follows:

- Continuous i.i.d. response (4 setups): We generate $n = 20$ samples from $y_i = x_i^\top \beta^\star + \epsilon_i$, $i = 1, \ldots, n$. The errors $\epsilon_i$ are either i.i.d. draws from $\mathcal{N}(0,1)$ or from Student-$t$ distributions with 5, 4, or 3 degrees of freedom. Note that $\theta(F^\star)$ coincides with $(0, \beta^{\star\top})$ in the case of i.i.d. $\mathcal{N}(0,1)$ errors.

- Continuous response with heteroscedastic errors (3 setups): We generate $n = 20$ samples from $y_i = x_i^\top \beta^\star + \epsilon_i$, $i = 1, \ldots, n$, where $\epsilon_i \sim \mathcal{N}(0, \sigma_i^2)$. The standard deviation of the error $\sigma_i$ is determined by the first coordinate of $x_i$:

$$\sigma_i = \begin{cases} s_{\text{left}}, & \text{if } x_{i1} < \hat{\xi}_{0.25}, \\ s_{\text{mid}}, & \text{if } \hat{\xi}_{0.25} \leq x_{i1} \leq \hat{\xi}_{0.75}, \\ 1, & \text{if } x_{i1} > \hat{\xi}_{0.75}, \end{cases}$$

where $\hat{\xi}_{0.25}$ and $\hat{\xi}_{0.75}$ are the empirical quartiles of $x_{11}, \ldots, x_{n1}$. The hyperparameters $(s_{\text{left}}, s_{\text{mid}})$ are ordered from weak to strong by their deviation from $\sigma = 1$: $s_1 = (0.25, 0.5)$, $s_2 = (0.05, 0.25)$, and $s_3 = (0.01, 0.1)$.

- Binary response (4 setups): We generate $n = 100$ samples from $y_i \mid x_i \overset{\text{iid}}{\sim} \text{Bernoulli}(L(x_i^\top \beta^\star))$, $i = 1, \ldots, n$, with different link functions $L$. The link is either the logistic function:

$$L(u) = (1 + \exp(-u))^{-1},$$

or the distribution function of a Gaussian-mixture model (GMM):

$$L(u) = 0.7 \, \Phi(u \mid a, 1) + 0.3 \, \Phi(u \mid 2, 1),$$

where $\Phi(\cdot \mid m, 1)$ denotes the distribution function of a Gaussian with mean $m$ and unit variance, and $a \in \{0, -1, -2\}$. Note that $\theta(F^\star)$ coincides with $(0, \beta^{\star\top})$ in the case of the logistic link.

**Real data.** We apply `train_test_split` in `scikit-learn` (Pedregosa et al., 2011) to the whole dataset, and use the training split as the dataset $z_{1:n}$. We repeat this procedure 100 times to generate repetitions of $z_{1:n}$. For datasets with highly skewed responses or features, we apply stratified sampling on these variables. For `yeast` and `wine`, which have highly skewed multiclass categorical responses, we also remove classes with fewer than 3% of observations. Multinomial logistic regression on a small dataset with an extremely skewed response remains difficult with our method, as the design matrix tends to be ill-conditioned when applying the Bayesian bootstrap on $x$.

The training sets are drawn from the population dataset with stratification on all categorical and numerical variables with few unique values. In certain datasets with heavily skewed numerical variables, we also stratify using bins of these numerical variables. The bin boundaries are determined to ensure that each bin contains approximately the same number of data points. The number of bins depends on the dataset and is chosen such that the marginal distribution of the training set roughly matches that of the population dataset.

The details of the 19 real-world datasets are provided in Tables 3 and 4.

*Table 3.* Details of the real-world datasets.

| $z_{1:n}$ | Data ID | $n$ | $p$ | $n/p$ | Population size | Num. classes | Num. cont. features | Num. cat. features |
|---|---|---|---|---|---|---|---|---|
| concrete | OpenML 44959 | 100 | 7 | 14.3 | 1030 | 0 | 8 | 0 |
| quake | OpenML 550 | 50 | 4 | 12.5 | 2178 | 0 | 3 | 0 |
| airfoil | OpenML 44957 | 50 | 5 | 10.0 | 1503 | 0 | 5 | 0 |
| energy | OpenML 44960 | 50 | 6 | 8.3 | 768 | 0 | 8 | 0 |
| fish | OpenML 44970 | 50 | 6 | 8.3 | 908 | 0 | 6 | 0 |
| kin8nm | OpenML 44980 | 50 | 9 | 5.6 | 8192 | 0 | 8 | 0 |
| auction | OpenML 44958 | 50 | 11 | 4.5 | 2043 | 0 | 5 | 2 |
| grid | OpenML 44973 | 50 | 12 | 4.2 | 10000 | 0 | 12 | 0 |
| abalone | OpenML 45042 | 20 | 5 | 4.0 | 4177 | 0 | 7 | 1 |
| rice | UCI 545 | 100 | 4 | 25.0 | 3810 | 2 | 7 | 0 |
| sepsis | OpenML 827 | 100 | 4 | 25.0 | 110341 | 2 | 2 | 1 |
| banknote | OpenML 1462 | 100 | 4 | 25.0 | 1372 | 2 | 4 | 0 |
| mozilla | OpenML 1046 | 100 | 5 | 20.0 | 15545 | 2 | 5 | 0 |
| skin | OpenML 1502 | 50 | 3 | 16.7 | 245057 | 2 | 3 | 0 |
| blood | OpenML 1464 | 50 | 4 | 12.5 | 748 | 2 | 4 | 0 |
| phoneme | OpenML 1489 | 50 | 6 | 8.3 | 5404 | 2 | 5 | 0 |
| telescope | UCI 159 | 50 | 9 | 5.6 | 19020 | 2 | 10 | 0 |
| yeast | OpenML 181 | 200 | 28 | 7.1 | 1350 | 5 | 8 | 0 |
| wine | OpenML 40498 | 200 | 40 | 5.0 | 4873 | 5 | 11 | 0 |

### E.4. Constructions of Credible Sets

**Joint credible sets.** Let $\widehat{\mu}$ and $\widehat{\Sigma}$ denote the empirical mean and covariance estimated from the posterior draws $\{\theta^{(l)}\}_{l=1}^{L}$. For martingale posteriors, we use a diagonal covariance structure $\widehat{\Sigma} = \text{diag}(\widehat{\sigma}_1^2, \ldots, \widehat{\sigma}_p^2)$ due to the limited number of posterior samples available for estimating a full covariance matrix. Our credible set is then defined as

$$\widehat{C}_{1-\alpha} = \left\{ \theta : (\theta - \widehat{\mu})^\top \widehat{\Sigma}^{-1} (\theta - \widehat{\mu}) \leq \widehat{r}_{1-\alpha}^2 \right\},$$

where

$$\widehat{r}_{1-\alpha}^2 = \text{quantile}_{1-\alpha} \{(\theta^{(l)} - \widehat{\mu})^\top \widehat{\Sigma}^{-1} (\theta^{(l)} - \widehat{\mu})\}_{l=1}^{L},$$

which is an empirical cutoff. This empirical-cutoff ellipsoid provides a data-driven approximation of the highest-posterior-density set without relying on $\chi_{p,1-\alpha}^2$, as stated in (12).

*Table 4.* Target name and the features included in the functional of interest.

| $z_{1:n}$ | Target Name | Dropped features in the functional |
|---|---|---|
| concrete | strength | blast_furnace_slag, water |
| quake | col_4 | |
| airfoil | sound_pressure | angle_of_attack |
| energy | heating_load | relative_compactness, surface_area, roof_area |
| fish | LC50 | MLOGP |
| kin8nm | y | |
| auction | verification.time | property.winner |
| grid | stab | p1 |
| abalone | Classnumberofrings | Length, Diameter, Whole_weight, Viscera_weight, Shell_weight |
| rice | Class | Area, Perimeter, Major_Axis_Length, Convex_Area |
| sepsis | hospital_outcome_1alive_0dead | |
| banknote | Class | V3 |
| mozilla | state | start |
| skin | Class | V2 |
| blood | Class | V3 |
| phoneme | Class | |
| telescope | class | fSize, fConc |
| yeast | class_protein_localization | erl, pox |
| wine | Class | V7, V8 |

**Marginal credible interval.** Let $\theta_j$ be the $j$-th dimension of $\theta$. The $(1 - \alpha)$ credible interval is computed from the empirical quantiles of its posterior draws:

$$\left[ \text{quantile}_{\alpha/2} \{\theta_j^{(l)}\}_{l=1}^L, \;\; \text{quantile}_{1-\alpha/2} \{\theta_j^{(l)}\}_{l=1}^L \right]. \tag{13}$$

## F. Diagnostics

Tracking each dimension of $\theta(F_N)$ as $N$ grows is a common approach for diagnosing the convergence of $\theta(F_N)$. However, this becomes impractical when $\theta$ is high-dimensional. Instead, we monitor convergence using the (scaled) expected $L_1$-norm between $\theta(F_n)$ and $\theta(F_N)$:

$$\mathbb{E}_{F_N} \left[ \frac{1}{p} \|\theta(F_n) - \theta(F_N)\|_1 \right],$$

where $p := \dim(\theta)$ and $\theta(F_n)$ is the empirical risk minimiser (or maximum likelihood estimate based on $z_{1:n}$ in our experiments). In practice, we simulate many realisations of $F_N$ to approximate this expectation. This expectation is shown in Figure 3 and corresponds to the black solid trajectories in Figure 6. In addition to the expected value, we also present the $L_1$-norm for each realisation of $F_N$ in Figure 6. While the paths corresponding to individual realisations of $F_N$ exhibit variability, they generally stabilise as $N$ increases, providing further empirical evidence that $\theta(F_N)$ converges.

For the slower-converging setups, we extend the diagnostic to longer rollouts runs in Figure 7. The selected setups include the `skin`, `yeast`, `wine`, and GMM classification experiments.

We also evaluate whether the convergence diagnostic is specific to the particular observed dataset $z_{1:n}$ used in Figure 3. Figure 8 shows the expected scaled $L_1$ trajectories over 20 independent realisations of $z_{1:n}$ for each setup. The trajectories generally stabilise, but a small number of trajectories spike or diverge. These edge cases typically correspond to realisations that lack diversity. As a result, TabPFN tends to produce predictions that lead to an ill-conditioned design matrix in the loss function (resulting in an unstable $\theta$).

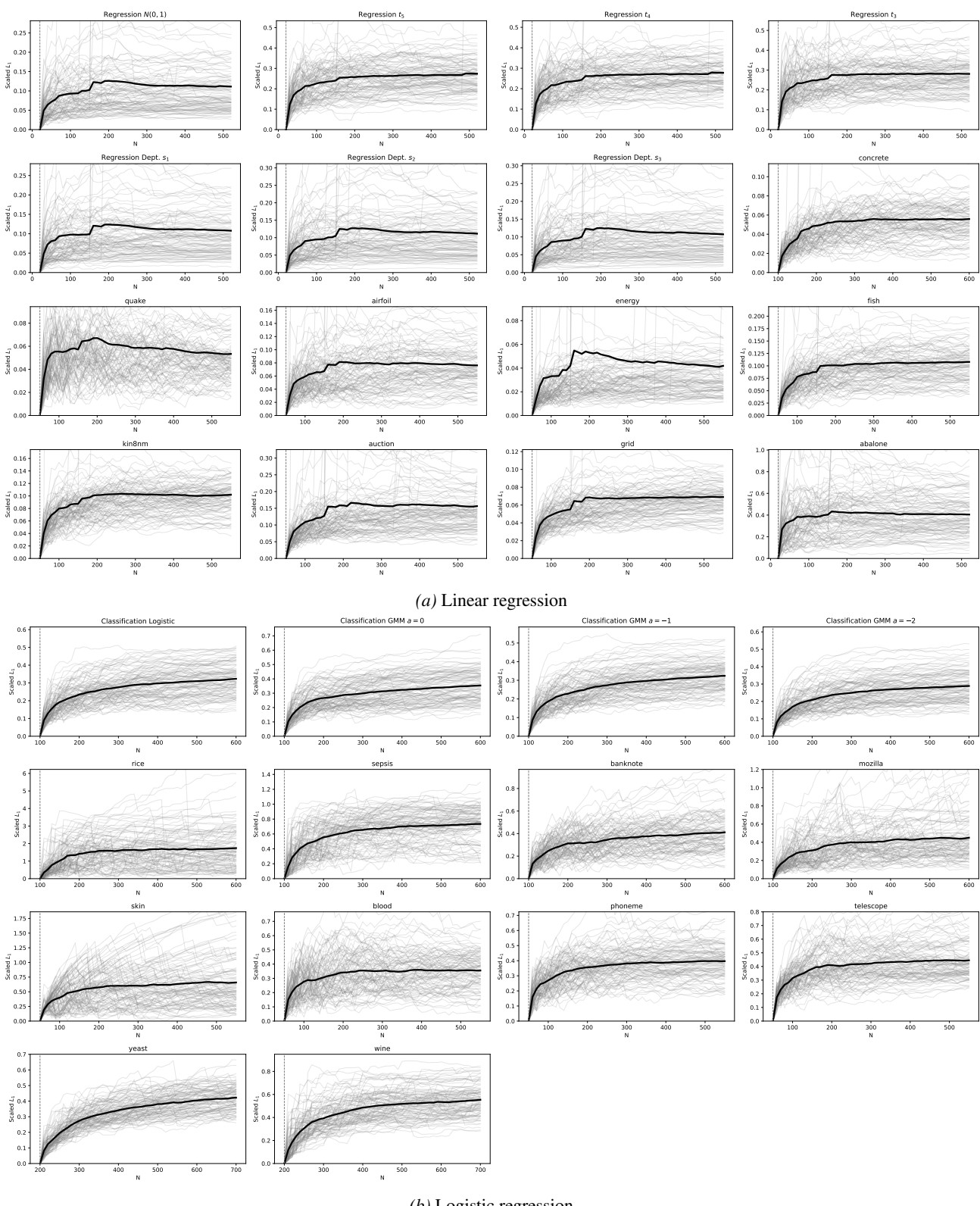

*(a)* Linear regression

*(b)* Logistic regression

*Figure 6.* $L_1$-norm between $\theta(F_n)$ and $\theta(F_N)$ as the rollout length $N$ increases. Each plot corresponds to a realisation $z_{1:n}$ of a particular setup. Each of the grey lines corresponds to a realisation of $F_N$, and the black solid lines are their average.

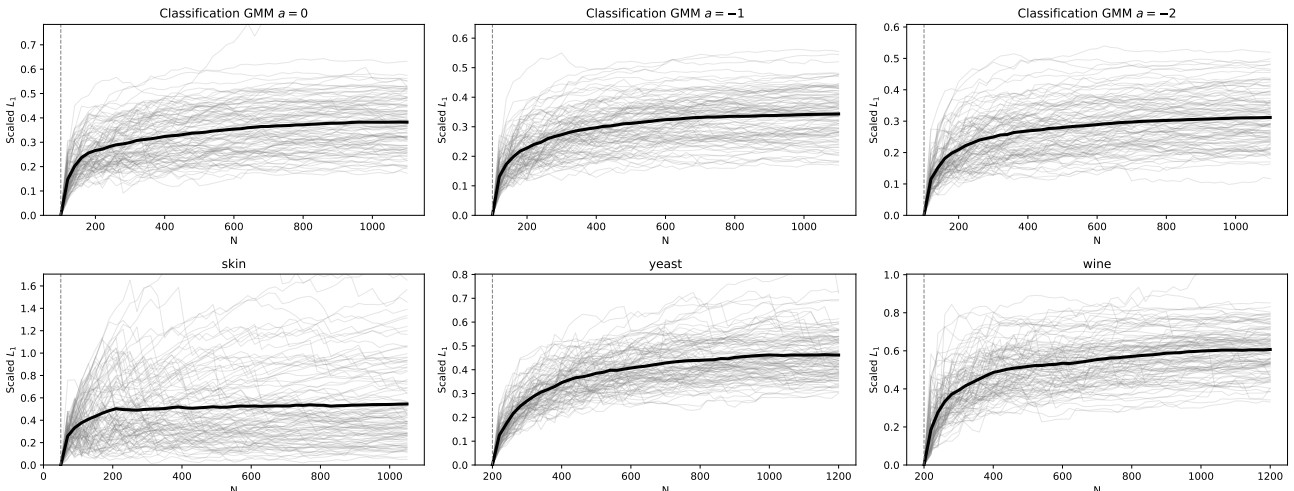

*Figure 7.* Scaled $L_1$-norm between $\theta(F_n)$ and $\theta(F_N)$ under longer TabMGP rollouts for the slower-converging setups. Each grey line corresponds to one rollout, and the black line is the average.

*Table 5.* Semi-synthetic MLP experiments. These setups combine real covariates with known nonlinear response mechanisms, allowing coverage to be evaluated against a known population target while preserving realistic tabular covariate structure. Reported values are coverage (Rate) and the median trace of the posterior covariance (Size), computed over 100 repetitions. Target coverage is 0.95.

| | TabMGP | | BB | | Copula | | Bayes | | Asymptotic | |
|---|---|---|---|---|---|---|---|---|---|---|
| Setup | Rate | Size | Rate | Size | Rate | Size | Rate | Size | Rate | Size |
| concrete-semi | 0.86 | 0.1465 | 0.84 | 0.1620 | 1.00 | 0.3580 | 0.94 | 0.2518 | 1.00 | 0.5189 |
| phoneme-semi | 0.98 | 4.1763 | 0.94 | 5.1099 | 1.00 | 10.5623 | 0.76 | 1.5573 | 1.00 | 3.1998 |

## G. Additional Coverage Experiments with Semi-Synthetic Setups

The real-data experiments preserve realistic covariate and response structure, but the population target must be approximated by the empirical distribution of the full dataset. Conversely, the synthetic experiments provide an exactly known data-generating process, but use simple covariate distributions. To combine these advantages, we include semi-synthetic setups that use real covariates together with a known nonlinear response mechanism. This allows us to evaluate coverage against a well-defined $\theta(F^\star)$ while retaining the covariate geometry of real tabular datasets.

We construct two such setups, one for continuous-response regression and one for binary classification. For Concrete, we use the covariates from OpenML 4353 and train a deterministic multi-layer perceptron (MLP) with hidden layers $(64, 32)$ and ReLU activations on the original compressive-strength response. The fitted network is then treated as the response-generating mechanism, producing continuous synthetic responses from the observed covariates. For Phoneme, we use the covariates from OpenML 1489 and train a deterministic binary MLP with hidden layers $(128, 64, 32)$, batch normalisation, ReLU activations, and dropout during training. Synthetic binary responses are generated by thresholding the fitted network probabilities at $0.5$. In both cases, repeated datasets are obtained by sampling covariates from the empirical covariate distribution and applying the fitted MLP response rule.

The corresponding coverage results are reported in Table 5. In these semi-synthetic setups, TabMGP remains competitive with BB while producing smaller credible sets, and Copula again tends to achieve coverage by using substantially larger sets. Bayes is well calibrated for Concrete but undercovers on Phoneme, whereas the asymptotic approximation overcovers in both setups.

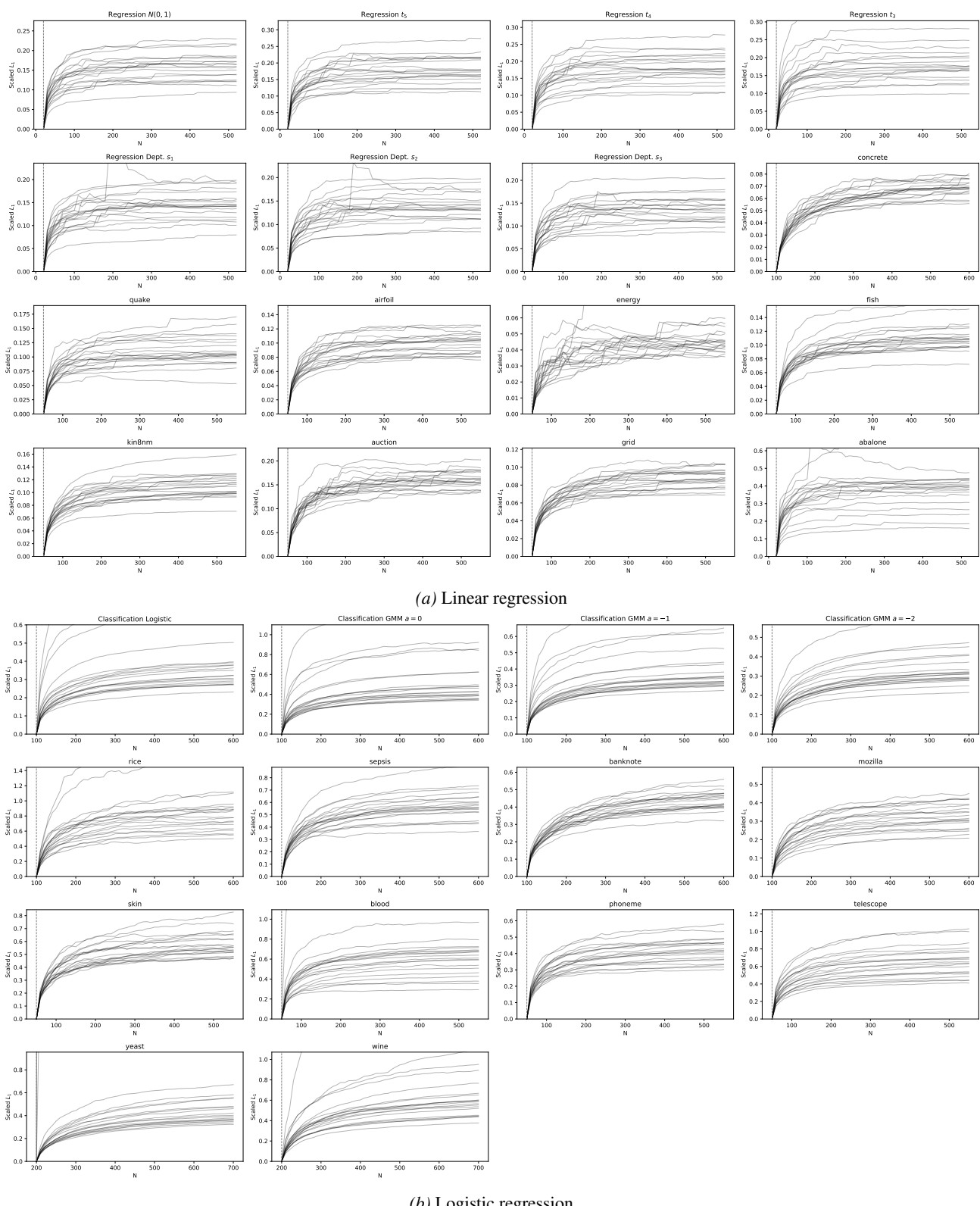

*(a)* Linear regression

*(b)* Logistic regression

*Figure 8.* Expected scaled $L_1$-norm between $\theta(F_n)$ and $\theta(F_N)$ for 20 independent realisations of $z_{1:n}$ in each setup. Most trajectories stabilise, while a few spiking trajectories indicate failure cases with low-diversity realisations of $z_{1:n}$.

*Table 6.* Coverage for linear regression, comparing TabMGP with standard Bayesian posteriors under Gaussian priors (Hessian and Diffuse) and copula-based methods with either standard initialisation (Fong et al., 2023) or TabPFN initialisation (Nagler & Rügamer, 2025). Reported values are coverage (Rate) and the median trace of the posterior covariance (Size), computed over 100 repetitions. The values of TabMGP, Bayes (Hessian) and Copula (Standard) are copied from Table 1 for ease of comparison. Target coverage is 0.95.

| Setup | TabMGP | | Bayes (Hessian) | | Bayes (Diffuse) | | Copula (Standard) | | Copula (TabPFN) | |
|---|---|---|---|---|---|---|---|---|---|---|
| | Rate | Size | Rate | Size | Rate | Size | Rate | Size | Rate | Size |
| $\mathcal{N}(0,1)$ | 1.00 | 0.45 | 1.00 | 0.65 | 1.00 | 1.29 | **0.99** | **0.35** | 0.99 | 0.46 |
| $t_5$ | 1.00 | 0.49 | 1.00 | 0.65 | 1.00 | 1.29 | **0.99** | **0.37** | 0.99 | 0.52 |
| $t_4$ | 0.99 | 0.51 | 0.99 | 0.65 | 1.00 | 1.30 | **0.98** | **0.37** | 0.99 | 0.53 |
| $t_3$ | 1.00 | 0.48 | 0.98 | 0.65 | 0.98 | 1.30 | **0.97** | **0.35** | 1.00 | 0.55 |
| $s_1$ | 1.00 | 0.37 | 1.00 | 0.65 | 1.00 | 1.30 | 1.00 | 0.36 | **1.00** | **0.32** |
| $s_2$ | 1.00 | 0.36 | 1.00 | 0.65 | 1.00 | 1.30 | 1.00 | 0.36 | **1.00** | **0.28** |
| $s_3$ | 1.00 | 0.33 | 1.00 | 0.65 | 1.00 | 1.30 | 1.00 | 0.37 | **1.00** | **0.26** |
| concrete | 0.91 | 0.06 | 0.87 | 0.05 | **1.00** | **0.10** | 1.00 | 0.12 | 0.67 | 0.04 |
| quake | **1.00** | **0.03** | 0.91 | 0.04 | 0.98 | 0.09 | 1.00 | 0.18 | 1.00 | 0.18 |
| airfoil | 0.96 | 0.08 | **0.96** | **0.06** | 1.00 | 0.12 | 0.97 | 0.11 | 0.88 | 0.07 |
| energy | **1.00** | **0.04** | 1.00 | 0.07 | 1.00 | 0.14 | 1.00 | 0.06 | 0.17 | 0.00 |
| fish | 0.90 | 0.10 | 0.91 | 0.08 | 1.00 | 0.16 | **0.95** | **0.10** | 0.94 | 0.14 |
| kin8nm | **0.95** | **0.13** | 0.84 | 0.11 | 0.99 | 0.22 | 0.63 | 0.08 | 0.99 | 0.17 |
| auction | **0.99** | **0.66** | 1.00 | 0.81 | 1.00 | 1.64 | 0.97 | 1.44 | 0.67 | 0.22 |
| grid | **0.97** | **0.13** | 0.97 | 0.16 | 1.00 | 0.31 | 0.88 | 0.10 | 0.98 | 0.14 |
| abalone | **0.97** | **0.95** | 0.84 | 0.81 | 0.98 | 1.62 | 0.94 | 0.98 | 0.87 | 0.78 |

## H. Additional Coverage Experiments with Alternative Posterior Constructions

We reconstruct the standard Bayes posterior as detailed in Appendix E.1 but replace the prior with a diffuse $\mathcal{N}(0, 10^2)$. We find that the resulting credible sets are excessively wide (Table 6 and Table 7) and thus exclude it from the main text to ensure a fair comparison with TabMGP.

We also compare TabMGP against the copula method initialised with TabPFN, as suggested in Nagler & Rügamer (2025). However, the results are mixed and we do not observe a substantial difference between the standard and TabPFN-initialised copula methods (Table 6 and Table 7).

## I. Sensitivity to the Rollout Length

We evaluate the sensitivity of the joint credible sets produced by TabMGP to the rollout length $N$. In our implementation, $N = n + T$, where $n$ is the number of observed data points and $T$ is the number of forward-sampling steps. We therefore vary $T$ while keeping the observed sample size fixed. We focus on the slower-converging settings: GMM, `skin`, `yeast`, and `wine`.

Figure 9 reports the results for $T$ ranging from 100 to 1000 forward-sampling steps. As $T$ increases, the estimated coverage and credible-set size should stabilise once the distribution of $\theta(F_N)$ is sufficiently close to its limiting behaviour. We find that most curves are largely stable by $T = 750$. The exception is a slight upward trend in credible-set size for `skin`. Together with Figure 8, this suggests that the rollout trajectories for some repetitions of $z_{1:n}$ have not yet converged. We also observe a drop in credible-set size for `wine` between $T = 750$ and $T = 1000$, which we attribute to numerical instability in some repetitions.

*Table 7.* Coverage for logistic regression, comparing TabMGP with standard Bayesian posteriors under Gaussian priors (Hessian and Diffuse) and copula-based methods with either standard initialisation (Fong et al., 2023) or TabPFN initialisation (Nagler & Rügamer, 2025). Reported values are coverage (Rate) and the median trace of the posterior covariance (Size), computed over 100 repetitions. The values of TabMGP, Bayes (Hessian) and Copula (Standard) are copied from Table 2 for ease of comparison. Target coverage is 0.95.

| | TabMGP | | Bayes (Hessian) | | Bayes (Diffuse) | | Copula (Standard) | | Copula (TabPFN) | |
| --- | --- | --- | --- | --- | --- | --- | --- | --- | --- | --- |
| Setup | Rate | Size | Rate | Size | Rate | Size | Rate | Size | Rate | Size |
| Logistic | 0.84 | 1.30 | 0.53 | 0.78 | **0.99** | **1.94** | 1.00 | 2.72 | 1.00 | 3.44 |
| GMM(0) | 0.91 | 2.44 | 0.60 | 1.34 | **0.99** | **3.66** | 1.00 | 4.01 | 1.00 | 5.75 |
| GMM(-1) | 0.83 | 1.27 | 0.58 | 0.79 | **1.00** | **1.95** | 1.00 | 2.64 | 1.00 | 3.33 |
| GMM(-2) | 0.78 | 0.88 | 0.52 | 0.57 | **1.00** | **1.37** | 1.00 | 2.08 | 1.00 | 2.60 |
| rice | **1.00** | **3.74** | 0.91 | 2.87 | 1.00 | 6.66 | 1.00 | 28.41 | 1.00 | 31.80 |
| sepsis | **0.95** | **0.69** | 0.98 | 1.18 | 1.00 | 3.23 | 1.00 | 10.00 | 1.00 | 10.80 |
| banknote | 0.99 | 1.16 | **0.99** | **0.64** | 1.00 | 1.48 | 1.00 | 1.53 | 1.00 | 1.54 |
| mozilla | **0.96** | **1.42** | 0.87 | 0.65 | 0.96 | 1.42 | 1.00 | 2.68 | 1.00 | 2.36 |
| skin | 1.00 | 1.39 | **1.00** | **0.57** | 1.00 | 1.49 | 1.00 | 1.80 | 1.00 | 1.78 |
| blood | 0.87 | 1.07 | 0.84 | 0.86 | **1.00** | **2.06** | 1.00 | 5.40 | 1.00 | 6.16 |
| phoneme | **1.00** | **1.32** | 0.73 | 0.83 | 1.00 | 2.15 | 1.00 | 5.62 | 1.00 | 6.92 |
| telescope | **0.99** | **5.96** | 0.67 | 3.50 | 1.00 | 9.39 | 1.00 | 50.69 | 1.00 | 61.30 |
| yeast | **0.97** | **11.10** | 0.33 | 10.82 | 1.00 | 26.90 | – | – | – | – |
| wine | 0.93 | 11.04 | 0.09 | 12.51 | **0.98** | **33.10** | – | – | – | – |

*Figure 9.* Coverage and credible-set size of TabMGP for selected slow-converging setups across several forward-sampling steps $T = N - n$. Coverage is evaluated over 100 repetitions of $z_{1:n}$. Size is the median trace of the posterior covariance. We expect the curves to stabilise as $T$ increases. The target coverage is 0.95.

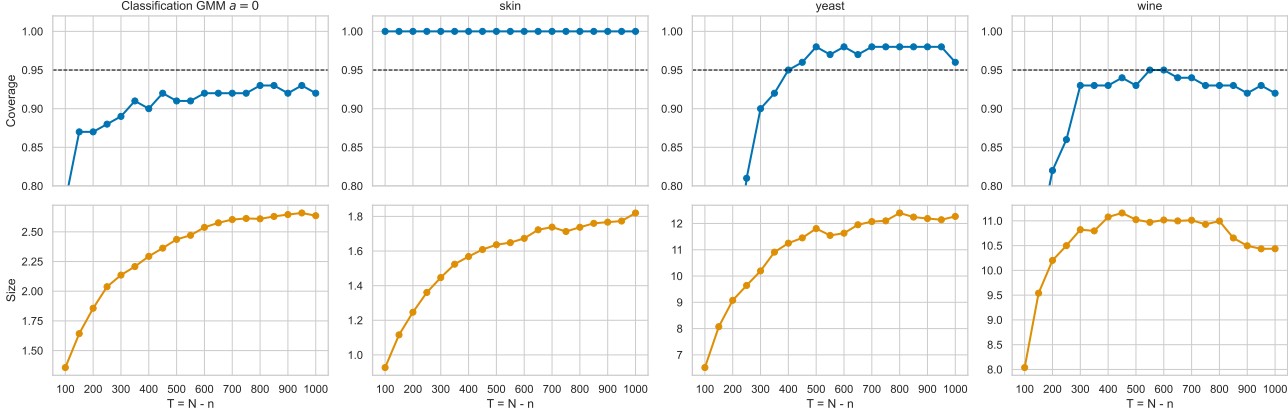

## J. Marginal Density and Credible Intervals of the Posteriors

We present density plots of all posterior constructions for a single repetition of $z_{1:n}$. 95% marginal credible intervals are shown as coloured horizontal bars in each plot. Due to space constraints, we display only the following setups:

1. Linear regression with synthetic i.i.d. Gaussian data (Figure 10);

2. Logistic regression with synthetic Bernoulli data with a logistic link (Figure 11);

3. Linear regression with the `abalone` dataset (Figure 12);

4. Logistic regression with the `telescope` dataset (Figure 13);

5. Multinomial logistic regression with the `yeast` dataset (Figure 14).

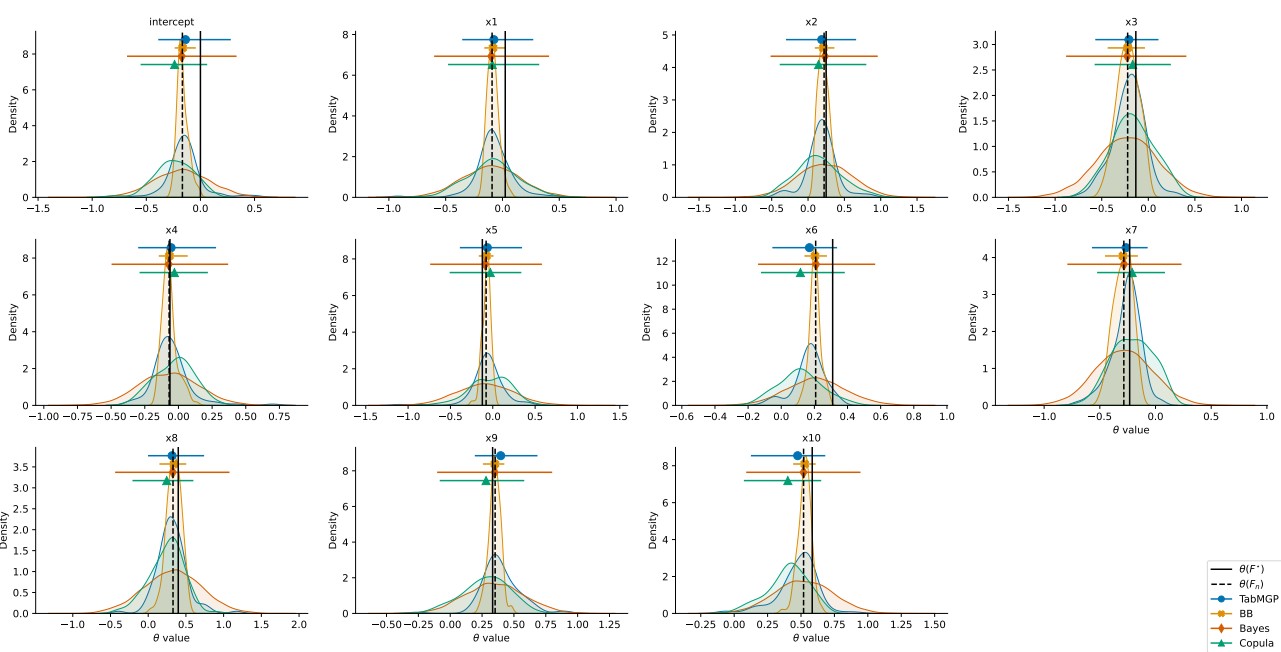

*Figure 10.* Marginal density plots for linear regression with synthetic i.i.d. Gaussian data. For each posterior density, the 95% marginal credible interval is shown as a horizontal bar, and the posterior mean is marked. The solid and dashed black vertical lines correspond to $\theta(F^\star)$ and $\theta(F_n)$, respectively.

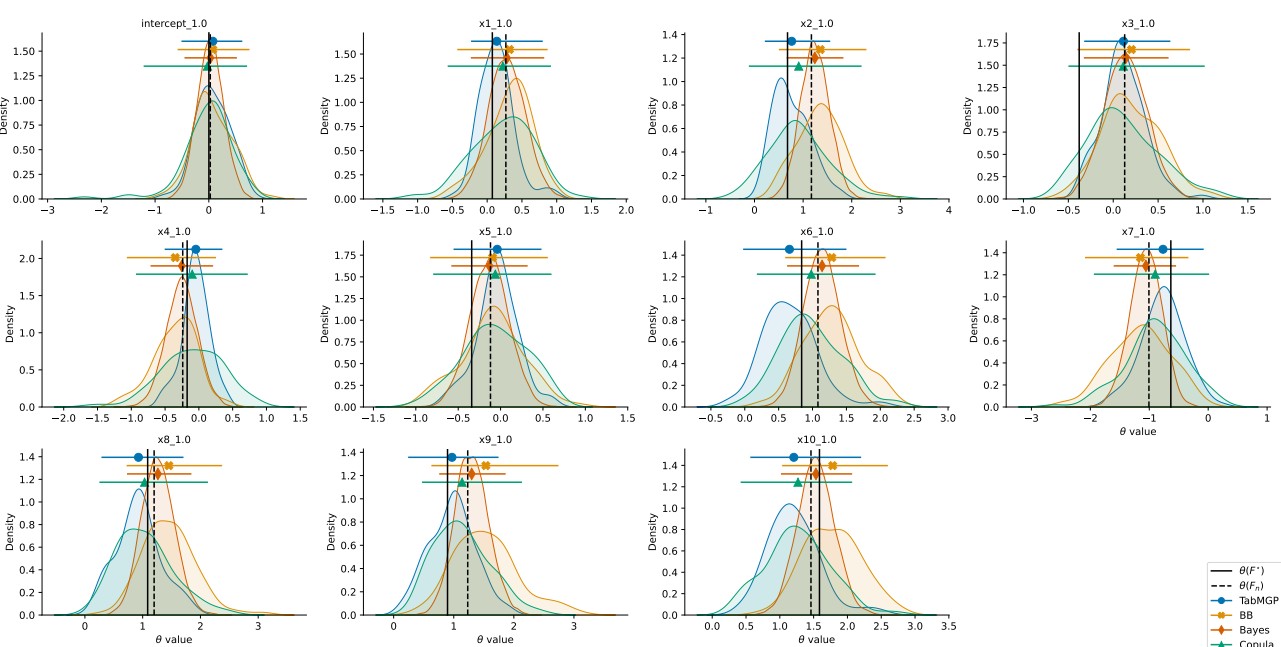

*Figure 11.* Marginal density plots for logistic regression with synthetic Bernoulli data and a logistic link. For each posterior density, the 95% marginal credible interval is shown as a horizontal bar, and the posterior mean is marked. The solid and dashed black vertical lines correspond to $\theta(F^\star)$ and $\theta(F_n)$, respectively.

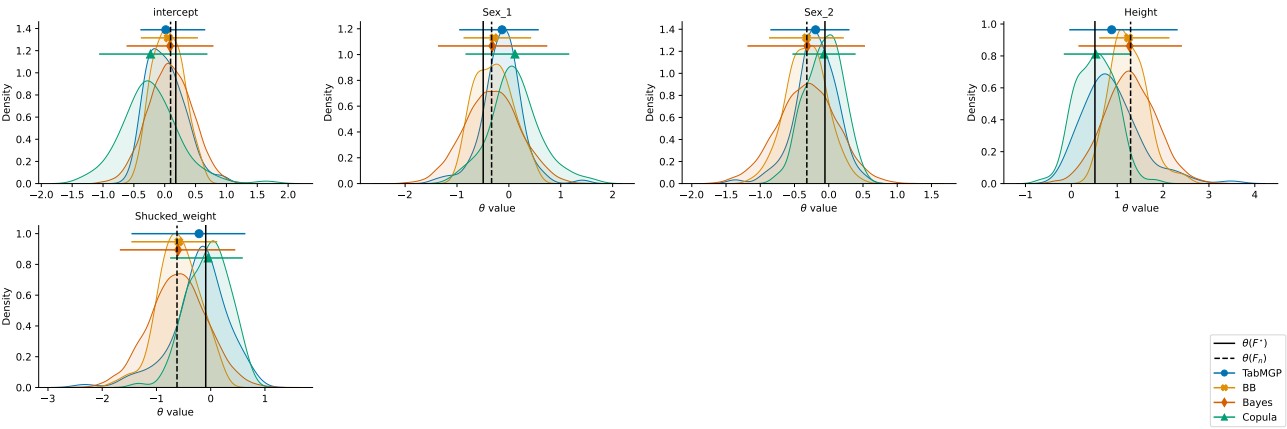

*Figure 12.* Marginal density plots for the `abalone` dataset. For each posterior density, the 95% marginal credible interval is shown as a horizontal bar, and the posterior mean is marked. The solid and dashed black vertical lines correspond to $\theta(F^\star)$ and $\theta(F_n)$, respectively.

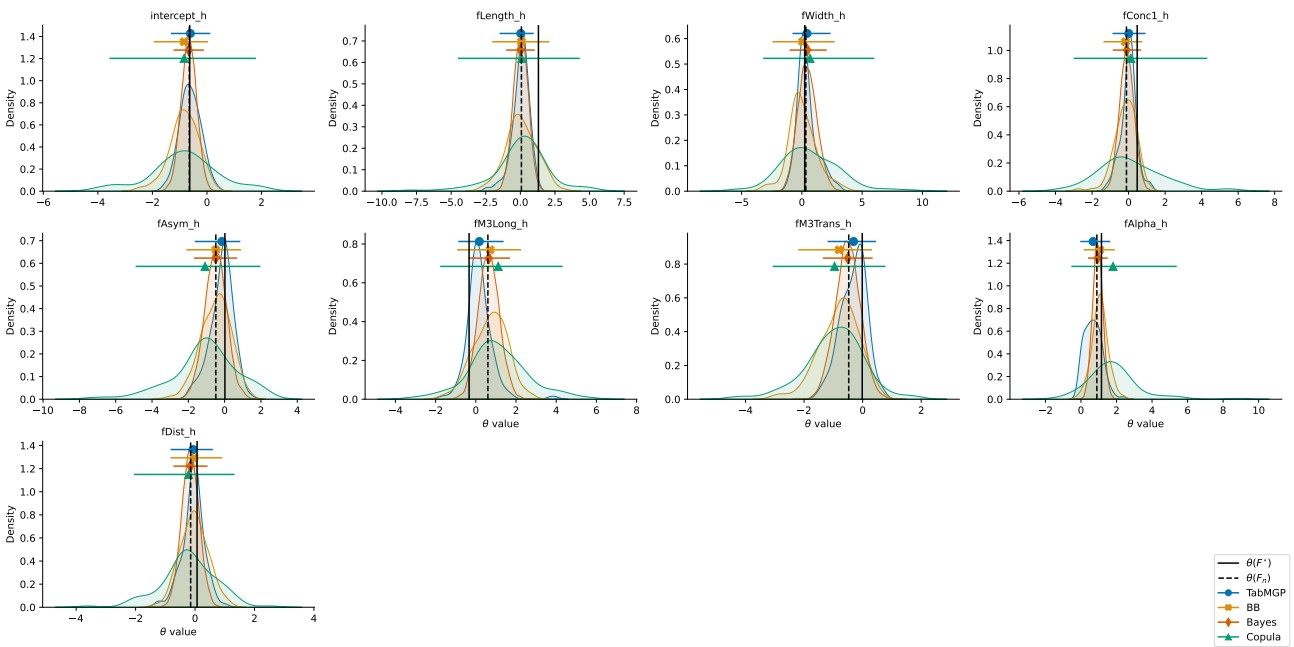

*Figure 13.* Marginal density plots for the `telescope` dataset. For each posterior density, the 95% marginal credible interval is shown as a horizontal bar, and the posterior mean is marked. The solid and dashed black vertical lines correspond to $\theta(F^\star)$ and $\theta(F_n)$, respectively.

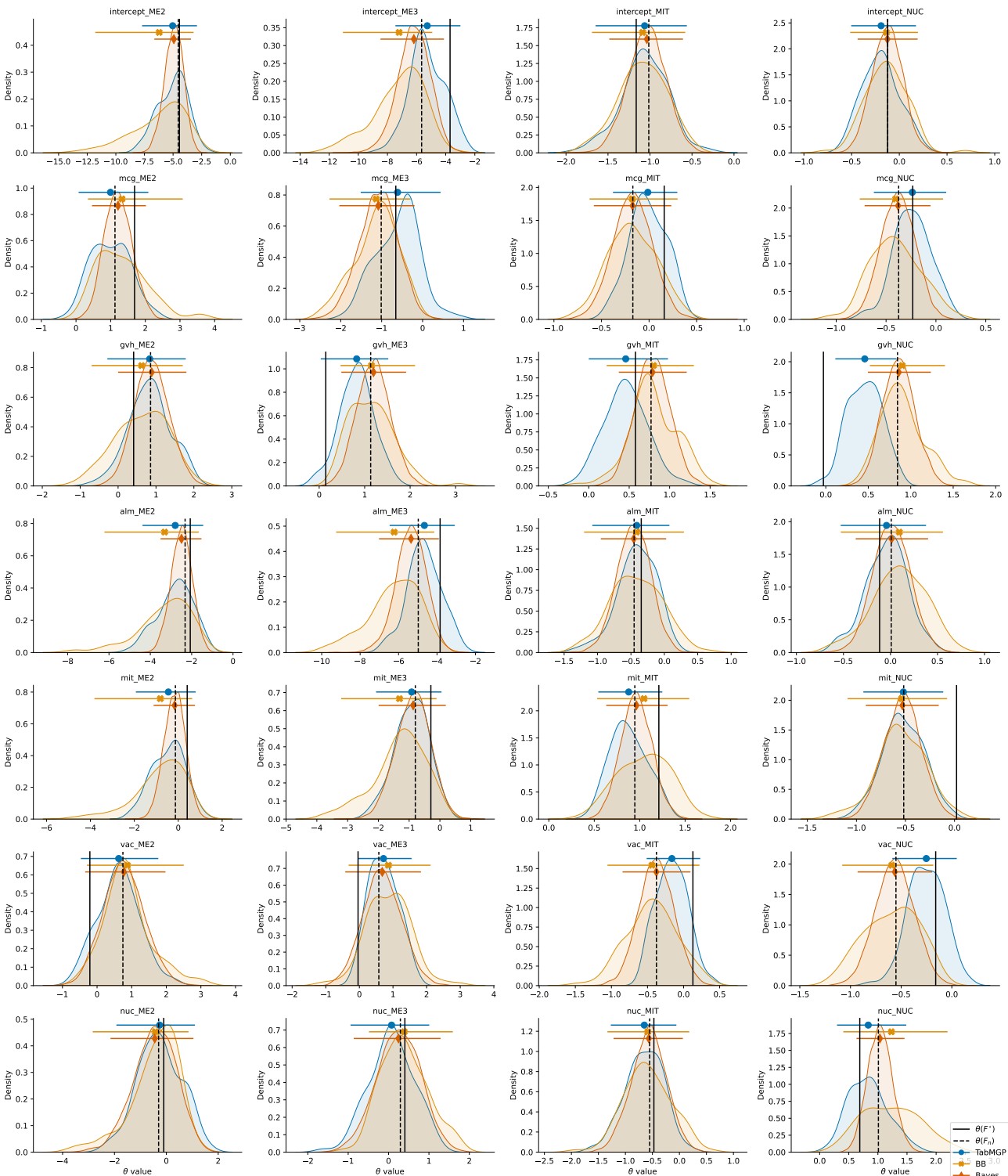

*Figure 14.* Marginal density plots for the `yeast` dataset. For each posterior density, the 95% marginal credible interval is shown as a horizontal bar, and the posterior mean is marked. The solid and dashed black vertical lines correspond to $\theta(F^\star)$ and $\theta(F_n)$, respectively.

## K. Coverage Rates of Marginal Credible Intervals

In addition to the joint credible sets discussed in the main text, we evaluate the performance of the $95\%$ marginal credible intervals (13) for each parameter. Figure 15 shows boxplots of the marginal coverage rates for each method across all parameters in our experiments. While most methods' coverage is centred near the nominal 0.95 level, BB exhibits substantial undercoverage. Bayes also tends to exhibit undercoverage in logistic regression.

The subsequent tables (Table 8 to Table 15) show the empirical coverage and the median width for each parameter, computed over 100 data realisations. The method achieving at least the target coverage with the smallest size is highlighted in bold. If none of the methods achieve the target coverage, the method with the highest coverage is highlighted in bold instead. Summarising across all 30 setups and 293 individual parameters, TabMGP is highlighted in bold 152 times, followed by Copula (100), the classical Bayesian posterior (37), and Bayesian bootstrap (4). TabMGP performs particularly well in multinomial logistic regression relative to the competing methods.

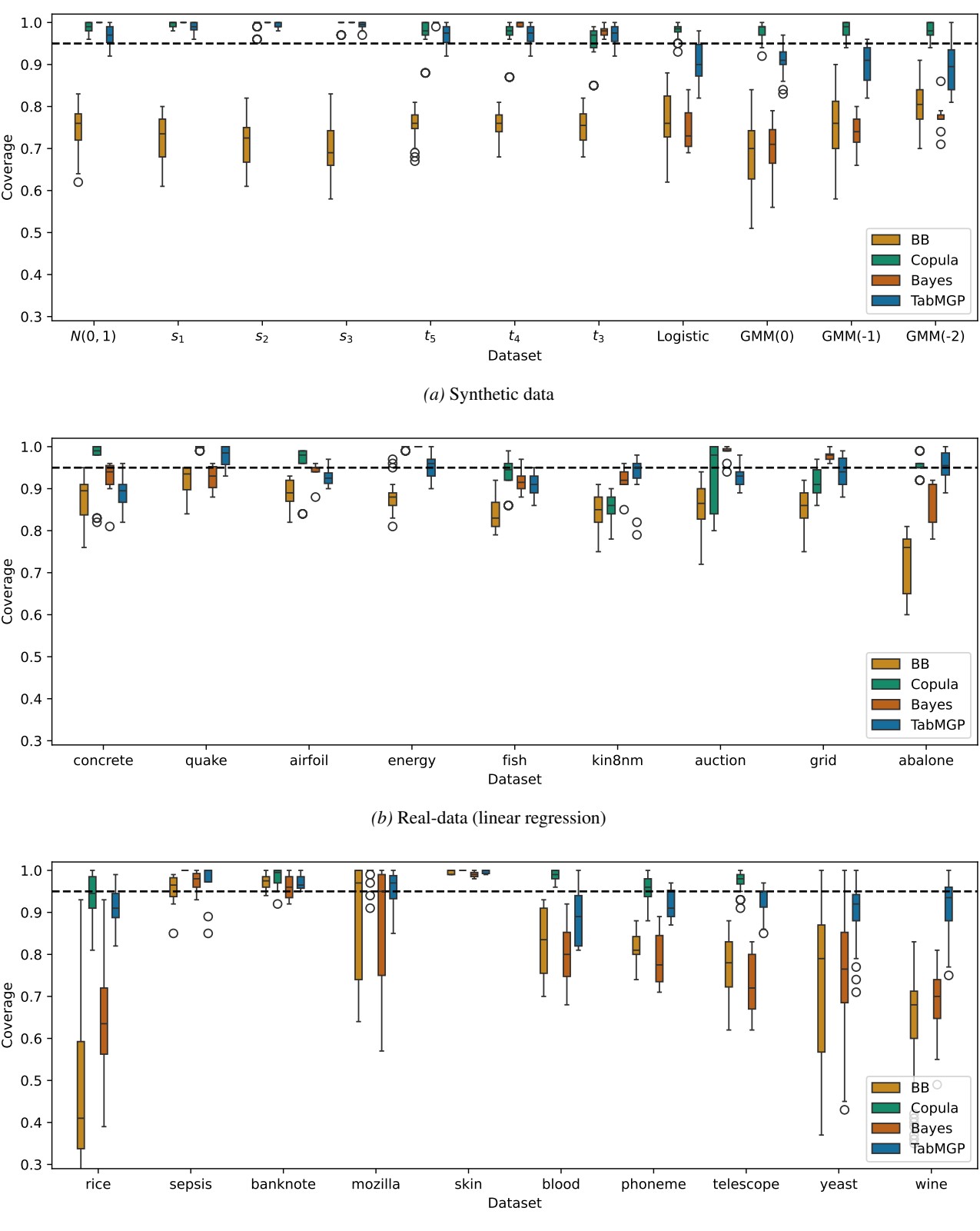

*(a)* Synthetic data

*(b)* Real-data (linear regression)

*(c)* Real-data (logistic regression)

*Figure 15.* Coverage of the 95% marginal credible intervals. Each data point in the boxplots represents the coverage for a single dimension of $\theta$ over 100 realisations of $z_{1:n}$.

*Table 8.* Coverage (rate) and median width (size) over 100 repetitions of marginal credible intervals for linear regression with synthetic i.i.d. data. The method achieving at least the target coverage with the smallest size is shown in bold. If none of the methods achieve the target coverage, the method with the highest coverage is shown in bold. The target coverage is 0.95.

| Setup | Feature Name | TabMGP Rate | TabMGP Size | BB Rate | BB Size | Copula Rate | Copula Size | Bayes Rate | Bayes Size |
|---|---|---|---|---|---|---|---|---|---|
| $\mathcal{N}(0,1)$ | intercept | 0.97 | 0.73 | 0.83 | 0.30 | **1.00** | **0.63** | 1.00 | 0.87 |
| | x1 | **1.00** | **0.64** | 0.77 | 0.31 | 1.00 | 0.67 | 1.00 | 0.94 |
| | x10 | 0.97 | 0.78 | 0.79 | 0.28 | **0.98** | **0.62** | 1.00 | 0.90 |
| | x2 | 0.94 | 0.65 | 0.75 | 0.30 | **0.99** | **0.65** | 1.00 | 0.88 |
| | x3 | **0.99** | **0.65** | 0.76 | 0.32 | 0.99 | 0.65 | 1.00 | 0.90 |
| | x4 | **0.99** | **0.63** | 0.81 | 0.29 | 0.99 | 0.64 | 1.00 | 0.90 |
| | x5 | **0.99** | **0.65** | 0.76 | 0.31 | 1.00 | 0.67 | 1.00 | 0.91 |
| | x6 | 0.95 | 0.68 | 0.72 | 0.31 | **0.97** | **0.65** | 1.00 | 0.92 |
| | x7 | 0.96 | 0.68 | 0.67 | 0.30 | **0.98** | **0.65** | 1.00 | 0.86 |
| | x8 | 0.94 | 0.69 | 0.78 | 0.30 | **0.96** | **0.66** | 1.00 | 0.90 |
| | x9 | 0.97 | 0.70 | 0.79 | 0.33 | **0.98** | **0.67** | 1.00 | 0.90 |
| $t_5$ | intercept | 0.97 | 0.78 | 0.79 | 0.34 | **1.00** | **0.65** | 1.00 | 0.87 |
| | x1 | **1.00** | **0.67** | 0.78 | 0.36 | 0.99 | 0.67 | 1.00 | 0.95 |
| | x10 | 0.92 | 0.84 | 0.76 | 0.37 | 0.88 | 0.63 | **1.00** | **0.90** |
| | x2 | 0.96 | 0.70 | 0.76 | 0.37 | **0.96** | **0.67** | 0.99 | 0.89 |
| | x3 | 0.99 | 0.66 | 0.76 | 0.36 | **1.00** | **0.65** | 1.00 | 0.91 |
| | x4 | 1.00 | 0.66 | 0.77 | 0.34 | **0.97** | **0.65** | 1.00 | 0.91 |
| | x5 | **0.99** | **0.66** | 0.75 | 0.35 | 0.99 | 0.68 | 1.00 | 0.91 |
| | x6 | 0.98 | 0.74 | 0.76 | 0.36 | **0.97** | **0.66** | 1.00 | 0.92 |
| | x7 | 0.99 | 0.69 | 0.78 | 0.36 | **1.00** | **0.66** | 1.00 | 0.87 |
| | x8 | 0.94 | 0.73 | 0.68 | 0.34 | **0.97** | **0.64** | 1.00 | 0.89 |
| | x9 | 0.97 | 0.74 | 0.81 | 0.34 | **0.98** | **0.66** | 1.00 | 0.90 |
| $t_4$ | intercept | 0.97 | 0.80 | 0.81 | 0.35 | **1.00** | **0.64** | 1.00 | 0.87 |
| | x1 | 1.00 | 0.68 | 0.79 | 0.38 | **0.99** | **0.67** | 1.00 | 0.95 |
| | x10 | 0.93 | 0.83 | 0.75 | 0.40 | 0.87 | 0.63 | **0.99** | **0.90** |
| | x2 | 0.94 | 0.68 | 0.76 | 0.39 | **0.96** | **0.66** | 0.99 | 0.88 |
| | x3 | 0.99 | 0.67 | 0.78 | 0.38 | **1.00** | **0.65** | 1.00 | 0.90 |
| | x4 | 1.00 | 0.68 | 0.78 | 0.36 | **0.98** | **0.65** | 1.00 | 0.92 |
| | x5 | 0.99 | 0.67 | 0.76 | 0.36 | **0.99** | **0.66** | 1.00 | 0.90 |
| | x6 | 0.99 | 0.75 | 0.75 | 0.38 | **0.97** | **0.67** | 1.00 | 0.91 |
| | x7 | 0.98 | 0.71 | 0.78 | 0.38 | **0.98** | **0.65** | 0.99 | 0.87 |
| | x8 | 0.94 | 0.71 | 0.70 | 0.34 | **0.97** | **0.64** | 0.99 | 0.89 |
| | x9 | 0.97 | 0.72 | 0.81 | 0.36 | **0.98** | **0.67** | 1.00 | 0.89 |
| $t_3$ | intercept | 0.97 | 0.80 | 0.82 | 0.38 | **0.96** | **0.62** | 0.99 | 0.87 |
| | x1 | 1.00 | 0.68 | 0.79 | 0.40 | **0.99** | **0.66** | 0.99 | 0.95 |
| | x10 | 0.93 | 0.83 | 0.74 | 0.41 | 0.85 | 0.62 | **0.98** | **0.91** |
| | x2 | 0.95 | 0.67 | 0.76 | 0.42 | **0.96** | **0.66** | 0.97 | 0.89 |
| | x3 | 0.99 | 0.67 | 0.78 | 0.40 | **0.98** | **0.65** | 0.98 | 0.90 |
| | x4 | 1.00 | 0.68 | 0.78 | 0.38 | **0.98** | **0.65** | 0.98 | 0.90 |
| | x5 | 0.99 | 0.66 | 0.73 | 0.39 | **0.98** | **0.66** | 1.00 | 0.91 |
| | x6 | **0.98** | **0.73** | 0.73 | 0.41 | 0.94 | 0.65 | 0.96 | 0.92 |
| | x7 | 0.98 | 0.69 | 0.77 | 0.40 | **0.98** | **0.64** | 0.98 | 0.87 |
| | x8 | 0.92 | 0.71 | 0.71 | 0.37 | 0.93 | 0.62 | **0.97** | **0.90** |
| | x9 | 0.98 | 0.73 | 0.79 | 0.38 | **0.97** | **0.66** | 0.97 | 0.90 |

*Table 9.* Coverage (rate) and median width (size) over 100 repetitions of marginal credible intervals for linear regression with synthetic, dependent-error data. The method achieving at least the target coverage with the smallest size is shown in bold. If none of the methods achieve the target coverage, the method with the highest coverage is shown in bold. The target coverage is 0.95.

| Setup | Feature Name | TabMGP | | BB | | Copula | | Bayes | |
|---|---|---|---|---|---|---|---|---|---|
| | | Rate | Size | Rate | Size | Rate | Size | Rate | Size |
| | intercept | 0.99 | 0.66 | 0.79 | 0.18 | **1.00** | **0.63** | 1.00 | 0.86 |
| | x1 | **1.00** | **0.55** | 0.73 | 0.19 | 1.00 | 0.67 | 1.00 | 0.94 |
| | x10 | 0.99 | 0.71 | 0.78 | 0.19 | **0.98** | **0.64** | 1.00 | 0.91 |
| | x2 | **0.97** | **0.61** | 0.73 | 0.18 | 1.00 | 0.65 | 1.00 | 0.89 |
| | x3 | **1.00** | **0.59** | 0.80 | 0.20 | 1.00 | 0.66 | 1.00 | 0.90 |
| $s_1$ | x4 | **1.00** | **0.57** | 0.77 | 0.19 | 1.00 | 0.66 | 1.00 | 0.91 |
| | x5 | **1.00** | **0.59** | 0.77 | 0.20 | 1.00 | 0.66 | 1.00 | 0.91 |
| | x6 | **0.98** | **0.64** | 0.67 | 0.19 | 0.99 | 0.65 | 1.00 | 0.92 |
| | x7 | **0.99** | **0.59** | 0.65 | 0.19 | 1.00 | 0.64 | 1.00 | 0.87 |
| | x8 | **0.97** | **0.63** | 0.72 | 0.19 | 0.98 | 0.66 | 1.00 | 0.89 |
| | x9 | **0.99** | **0.62** | 0.78 | 0.20 | 1.00 | 0.66 | 1.00 | 0.89 |
| | intercept | 1.00 | 0.65 | 0.82 | 0.14 | **1.00** | **0.63** | 1.00 | 0.86 |
| | x1 | **0.99** | **0.54** | 0.62 | 0.16 | 1.00 | 0.66 | 1.00 | 0.95 |
| | x10 | 0.99 | 0.70 | 0.75 | 0.15 | **0.96** | **0.64** | 1.00 | 0.91 |
| | x2 | **0.99** | **0.58** | 0.78 | 0.14 | 1.00 | 0.65 | 1.00 | 0.89 |
| | x3 | **1.00** | **0.55** | 0.72 | 0.15 | 1.00 | 0.65 | 1.00 | 0.90 |
| $s_2$ | x4 | **1.00** | **0.55** | 0.76 | 0.15 | 1.00 | 0.67 | 1.00 | 0.91 |
| | x5 | **0.99** | **0.58** | 0.76 | 0.16 | 1.00 | 0.67 | 1.00 | 0.91 |
| | x6 | **1.00** | **0.61** | 0.64 | 0.15 | 0.99 | 0.65 | 1.00 | 0.92 |
| | x7 | **0.99** | **0.57** | 0.75 | 0.15 | 1.00 | 0.63 | 1.00 | 0.87 |
| | x8 | **0.98** | **0.61** | 0.66 | 0.15 | 1.00 | 0.66 | 1.00 | 0.89 |
| | x9 | **1.00** | **0.60** | 0.77 | 0.16 | 1.00 | 0.65 | 1.00 | 0.90 |
| | intercept | **1.00** | **0.63** | 0.82 | 0.13 | 1.00 | 0.64 | 1.00 | 0.87 |
| | x1 | **0.99** | **0.55** | 0.62 | 0.15 | 1.00 | 0.66 | 1.00 | 0.95 |
| | x10 | 0.99 | 0.70 | 0.74 | 0.14 | **0.97** | **0.64** | 1.00 | 0.90 |
| | x2 | **0.98** | **0.58** | 0.78 | 0.13 | 1.00 | 0.65 | 1.00 | 0.89 |
| | x3 | **0.99** | **0.56** | 0.68 | 0.15 | 1.00 | 0.66 | 1.00 | 0.91 |
| $s_3$ | x4 | **1.00** | **0.57** | 0.69 | 0.14 | 1.00 | 0.67 | 1.00 | 0.91 |
| | x5 | **1.00** | **0.58** | 0.75 | 0.15 | 1.00 | 0.67 | 1.00 | 0.90 |
| | x6 | **1.00** | **0.61** | 0.62 | 0.15 | 1.00 | 0.65 | 1.00 | 0.92 |
| | x7 | **1.00** | **0.56** | 0.76 | 0.14 | 1.00 | 0.63 | 1.00 | 0.87 |
| | x8 | **0.98** | **0.60** | 0.67 | 0.14 | 1.00 | 0.66 | 1.00 | 0.90 |
| | x9 | **1.00** | **0.61** | 0.74 | 0.15 | 1.00 | 0.66 | 1.00 | 0.89 |

*Table 10.* Coverage (rate) and median width (size) over 100 repetitions of marginal credible intervals for logistic regression with synthetic data. The method achieving at least the target coverage with the smallest size is shown in bold. If none of the methods achieve the target coverage, the method with the highest coverage is shown in bold. The target coverage is 0.95.

| Setup | Feature Name | TabMGP | | BB | | Copula | | Bayes | |
|---|---|---|---|---|---|---|---|---|---|
| | | Rate | Size | Rate | Size | Rate | Size | Rate | Size |
| Logistic | intercept_1.0 | **0.96** | **1.11** | 0.88 | 1.36 | 0.98 | 1.67 | 0.84 | 0.93 |
| | x10_1.0 | 0.86 | 1.76 | 0.66 | 2.07 | **0.99** | **2.24** | 0.69 | 1.25 |
| | x1_1.0 | **0.97** | **0.97** | 0.86 | 1.44 | 0.99 | 1.69 | 0.77 | 0.95 |
| | x2_1.0 | 0.85 | 1.19 | 0.76 | 1.56 | **0.98** | **1.82** | 0.71 | 1.01 |
| | x3_1.0 | **0.95** | **1.04** | 0.81 | 1.41 | 1.00 | 1.69 | 0.80 | 0.97 |
| | x4_1.0 | **0.98** | **0.99** | 0.85 | 1.41 | 0.99 | 1.71 | 0.73 | 0.94 |
| | x5_1.0 | 0.94 | 1.04 | 0.86 | 1.44 | **0.99** | **1.73** | 0.81 | 0.95 |
| | x6_1.0 | 0.90 | 1.34 | 0.77 | 1.56 | **1.00** | **1.88** | 0.75 | 1.04 |
| | x7_1.0 | 0.89 | 1.21 | 0.79 | 1.54 | **1.00** | **1.85** | 0.72 | 1.00 |
| | x8_1.0 | 0.85 | 1.48 | 0.75 | 1.74 | **0.99** | **2.01** | 0.70 | 1.11 |
| | x9_1.0 | 0.89 | 1.44 | 0.73 | 1.67 | **0.99** | **1.94** | 0.69 | 1.08 |
| GMM(0) | intercept_1.0 | 0.90 | 1.48 | 0.73 | 2.02 | **0.99** | **2.16** | 0.70 | 1.24 |
| | x10_1.0 | 0.91 | 2.53 | 0.52 | 3.37 | **1.00** | **3.00** | 0.56 | 1.87 |
| | x1_1.0 | **0.96** | **1.21** | 0.82 | 1.84 | 1.00 | 1.98 | 0.74 | 1.14 |
| | x2_1.0 | 0.86 | 1.61 | 0.70 | 2.09 | **0.97** | **2.16** | 0.71 | 1.27 |
| | x3_1.0 | 0.91 | 1.33 | 0.79 | 1.79 | **0.99** | **1.96** | 0.75 | 1.16 |
| | x4_1.0 | **0.97** | **1.26** | 0.79 | 1.77 | 0.99 | 2.00 | 0.73 | 1.16 |
| | x5_1.0 | 0.94 | 1.24 | 0.84 | 1.76 | **0.97** | **1.89** | 0.79 | 1.16 |
| | x6_1.0 | 0.91 | 1.86 | 0.68 | 2.37 | **1.00** | **2.37** | 0.67 | 1.35 |
| | x7_1.0 | 0.93 | 1.60 | 0.74 | 2.02 | **0.99** | **2.11** | 0.75 | 1.26 |
| | x8_1.0 | 0.90 | 2.07 | 0.62 | 2.76 | **0.99** | **2.51** | 0.63 | 1.50 |
| | x9_1.0 | **0.95** | **1.92** | 0.64 | 2.54 | 1.00 | 2.33 | 0.66 | 1.43 |
| GMM(−1) | intercept_1.0 | **0.96** | **1.10** | 0.89 | 1.36 | 0.98 | 1.67 | 0.80 | 0.94 |
| | x10_1.0 | 0.90 | 1.78 | 0.66 | 1.96 | **1.00** | **2.25** | 0.71 | 1.26 |
| | x1_1.0 | **0.96** | **0.96** | 0.82 | 1.46 | 1.00 | 1.68 | 0.76 | 0.96 |
| | x2_1.0 | 0.85 | 1.24 | 0.77 | 1.54 | **0.99** | **1.80** | 0.76 | 1.02 |
| | x3_1.0 | 0.91 | 1.06 | 0.83 | 1.43 | **0.99** | **1.68** | 0.78 | 0.97 |
| | x4_1.0 | **0.96** | **0.99** | 0.82 | 1.39 | 0.99 | 1.69 | 0.74 | 0.95 |
| | x5_1.0 | 0.93 | 1.02 | 0.88 | 1.42 | **1.00** | **1.71** | 0.79 | 0.95 |
| | x6_1.0 | 0.88 | 1.36 | 0.75 | 1.59 | **1.00** | **1.86** | 0.72 | 1.04 |
| | x7_1.0 | 0.86 | 1.18 | 0.78 | 1.55 | **1.00** | **1.82** | 0.74 | 0.98 |
| | x8_1.0 | 0.86 | 1.51 | 0.68 | 1.81 | **0.99** | **2.02** | 0.66 | 1.11 |
| | x9_1.0 | 0.94 | 1.46 | 0.73 | 1.65 | **1.00** | **1.92** | 0.66 | 1.07 |
| GMM(−2) | intercept_1.0 | 0.92 | 1.05 | 0.84 | 1.35 | **0.98** | **1.67** | 0.79 | 0.89 |
| | x10_1.0 | 0.85 | 1.42 | 0.73 | 1.48 | **1.00** | **1.82** | 0.77 | 1.03 |
| | x1_1.0 | **0.97** | **0.81** | 0.89 | 1.22 | 0.97 | 1.56 | 0.86 | 0.85 |
| | x2_1.0 | 0.87 | 0.94 | 0.78 | 1.28 | **0.99** | **1.60** | 0.77 | 0.89 |
| | x3_1.0 | 0.94 | 0.85 | 0.88 | 1.24 | **0.97** | **1.58** | 0.78 | 0.86 |
| | x4_1.0 | **0.99** | **0.84** | 0.91 | 1.21 | 1.00 | 1.54 | 0.78 | 0.85 |
| | x5_1.0 | 0.90 | 0.88 | 0.82 | 1.26 | **1.00** | **1.53** | 0.78 | 0.86 |
| | x6_1.0 | 0.91 | 1.05 | 0.84 | 1.33 | **1.00** | **1.67** | 0.77 | 0.91 |
| | x7_1.0 | 0.83 | 0.97 | 0.84 | 1.33 | **1.00** | **1.61** | 0.77 | 0.87 |
| | x8_1.0 | 0.81 | 1.18 | 0.78 | 1.37 | **1.00** | **1.74** | 0.74 | 0.92 |
| | x9_1.0 | 0.83 | 1.08 | 0.79 | 1.33 | **1.00** | **1.65** | 0.71 | 0.91 |

*Table 11.* Coverage (rate) and median width (size) over 100 repetitions of marginal credible intervals for linear regression with real data. The method achieving at least the target coverage with the smallest size is shown in bold. If none of the methods achieve the target coverage, the method with the highest coverage is shown in bold. The target coverage is 0.95.

| Setup | Feature Name | TabMGP | | BB | | Copula | | Bayes | |
|---|---|---|---|---|---|---|---|---|---|
| | | Rate | Size | Rate | Size | Rate | Size | Rate | Size |
| concrete | age | 0.83 | 0.36 | 0.83 | 0.35 | **0.83** | **0.32** | 0.81 | 0.29 |
| | cement | 0.90 | 0.32 | 0.91 | 0.30 | **0.98** | **0.50** | 0.92 | 0.34 |
| | coarse_aggregate | 0.91 | 0.29 | 0.91 | 0.27 | **0.99** | **0.47** | 0.94 | 0.30 |
| | fine_aggregate | 0.95 | 0.29 | **0.95** | **0.26** | 1.00 | 0.45 | 0.96 | 0.31 |
| | fly_ash | **0.96** | **0.34** | 0.92 | 0.32 | 0.99 | 0.63 | 0.96 | 0.36 |
| | intercept | 0.91 | 0.29 | 0.92 | 0.26 | 1.00 | 0.55 | **0.95** | **0.28** |
| | superplasticizer | 0.91 | 0.37 | 0.85 | 0.35 | **0.99** | **0.49** | 0.90 | 0.36 |
| quake | col_1 | **1.00** | **0.23** | 0.88 | 0.44 | 0.99 | 0.50 | 0.95 | 0.45 |
| | col_2 | 0.93 | 0.25 | 0.94 | 0.51 | **1.00** | **0.80** | 0.88 | 0.41 |
| | col_3 | **1.00** | **0.24** | 0.95 | 0.50 | 1.00 | 0.81 | 0.91 | 0.40 |
| | intercept | 0.96 | 0.47 | 0.95 | 0.50 | 1.00 | 1.02 | **0.96** | **0.39** |
| airfoil | chord_length | 0.96 | 0.44 | 0.91 | 0.38 | 0.98 | 0.55 | **0.96** | **0.42** |
| | displacement_thickness | 0.93 | 0.45 | 0.87 | 0.40 | **0.96** | **0.59** | 0.94 | 0.42 |
| | free_stream_velocity | 0.94 | 0.40 | 0.93 | 0.35 | 0.99 | 0.55 | **0.95** | **0.41** |
| | frequency | **0.90** | **0.48** | 0.87 | 0.43 | 0.84 | 0.47 | 0.88 | 0.42 |
| | intercept | 0.93 | 0.42 | 0.93 | 0.36 | 0.99 | 0.63 | **0.95** | **0.40** |
| energy | glazing_area | 0.93 | 0.25 | 0.86 | 0.15 | **1.00** | **0.34** | 1.00 | 0.42 |
| | glazing_area_distribution | **0.96** | **0.25** | 0.89 | 0.15 | 0.99 | 0.33 | 1.00 | 0.43 |
| | intercept | 0.92 | 0.28 | 0.88 | 0.15 | 1.00 | 0.45 | **1.00** | **0.40** |
| | orientation | **0.96** | **0.24** | 0.88 | 0.15 | 1.00 | 0.34 | 1.00 | 0.41 |
| | overall_height | **0.97** | **0.26** | 0.89 | 0.15 | 1.00 | 0.49 | 1.00 | 0.42 |
| | wall_area | 1.00 | 0.25 | **0.96** | **0.12** | 1.00 | 0.35 | 1.00 | 0.43 |
| fish | CIC0 | 0.89 | 0.46 | 0.86 | 0.43 | 0.86 | 0.49 | **0.90** | **0.43** |
| | GATS1i | 0.92 | 0.41 | 0.92 | 0.34 | 0.96 | 0.46 | **0.97** | **0.42** |
| | NdsCH | 0.95 | 0.45 | 0.81 | 0.36 | **0.95** | **0.45** | 0.93 | 0.45 |
| | NdssC | **0.94** | **0.46** | 0.89 | 0.44 | 0.94 | 0.47 | 0.93 | 0.48 |
| | SM1_Dz | 0.89 | 0.48 | 0.83 | 0.38 | **0.92** | **0.48** | 0.88 | 0.45 |
| | intercept | 0.93 | 0.43 | 0.83 | 0.35 | **0.99** | **0.54** | 0.90 | 0.41 |
| kin8nm | intercept | 0.94 | 0.47 | 0.91 | 0.38 | 0.90 | 0.35 | **0.94** | **0.42** |
| | theta1 | **0.96** | **0.41** | 0.85 | 0.38 | 0.86 | 0.36 | 0.92 | 0.44 |
| | theta2 | **0.98** | **0.40** | 0.86 | 0.37 | 0.88 | 0.35 | 0.96 | 0.43 |
| | theta3 | **0.96** | **0.52** | 0.90 | 0.36 | 0.78 | 0.34 | 0.94 | 0.43 |
| | theta4 | **0.96** | **0.42** | 0.90 | 0.39 | 0.90 | 0.34 | 0.93 | 0.43 |
| | theta5 | 0.82 | 0.44 | 0.87 | 0.40 | 0.86 | 0.33 | **0.91** | **0.42** |
| | theta6 | **0.92** | **0.43** | 0.79 | 0.39 | 0.78 | 0.34 | 0.85 | 0.43 |
| | theta7 | **0.94** | **0.42** | 0.84 | 0.38 | 0.84 | 0.35 | 0.91 | 0.43 |
| | theta8 | **0.96** | **0.41** | 0.84 | 0.37 | 0.87 | 0.35 | 0.92 | 0.42 |

*Table 12.* Coverage (rate) and median width (size) over 100 repetitions of marginal credible intervals for linear regression with real data. The method achieving at least the target coverage with the smallest size is shown in bold. If none of the methods achieve the target coverage, the method with the highest coverage is shown in bold. The target coverage is 0.95.

| Setup | Feature Name | TabMGP | | BB | | Copula | | Bayes | |
|---|---|---|---|---|---|---|---|---|---|
| | | Rate | Size | Rate | Size | Rate | Size | Rate | Size |
| | intercept | 0.93 | 0.74 | 0.89 | 0.53 | 1.00 | 1.70 | **0.99** | **0.87** |
| | process.b1.capacity | **0.98** | **0.50** | 0.94 | 0.40 | 1.00 | 1.04 | 0.99 | 0.51 |
| | process.b2.capacity | 0.90 | 0.39 | 0.80 | 0.32 | 1.00 | 0.84 | **0.96** | **0.46** |
| | process.b3.capacity | 0.92 | 0.33 | 0.94 | 0.26 | 0.98 | 0.66 | **1.00** | **0.48** |
| | process.b4.capacity | 0.93 | 0.40 | 0.86 | 0.32 | 1.00 | 1.20 | **0.99** | **0.45** |
| auction | property.price | 0.91 | 0.45 | 0.81 | 0.37 | 1.00 | 1.03 | **0.99** | **0.52** |
| | property.product_2 | 0.91 | 1.39 | 0.87 | 1.08 | 0.84 | 1.61 | **0.94** | **1.26** |
| | property.product_3 | 0.93 | 1.06 | 0.89 | 0.85 | 0.80 | 1.61 | **0.99** | **1.48** |
| | property.product_4 | **0.96** | **1.04** | 0.89 | 0.84 | 0.88 | 1.39 | 1.00 | 1.44 |
| | property.product_5 | 0.94 | 1.14 | 0.89 | 0.89 | 0.97 | 1.67 | **1.00** | **1.64** |
| | property.product_6 | 0.94 | 0.99 | 0.93 | 0.77 | 0.82 | 1.50 | **0.99** | **1.34** |
| | g1 | 0.92 | 0.39 | 0.82 | 0.30 | 0.86 | 0.34 | **0.97** | **0.45** |
| | g2 | **0.98** | **0.40** | 0.88 | 0.30 | 0.94 | 0.33 | 1.00 | 0.45 |
| | g3 | **0.96** | **0.39** | 0.86 | 0.30 | 0.89 | 0.33 | 0.97 | 0.45 |
| | g4 | **0.98** | **0.40** | 0.92 | 0.31 | 0.93 | 0.33 | 0.97 | 0.45 |
| | intercept | 0.92 | 0.40 | 0.91 | 0.30 | 0.86 | 0.33 | **0.98** | **0.44** |
| grid | p2 | 0.97 | 0.36 | 0.90 | 0.29 | **0.96** | **0.34** | 0.98 | 0.45 |
| | p3 | 0.99 | 0.36 | 0.91 | 0.29 | **0.97** | **0.34** | 0.99 | 0.44 |
| | p4 | 0.99 | 0.38 | 0.87 | 0.29 | **0.97** | **0.34** | 0.98 | 0.44 |
| | tau1 | 0.91 | 0.39 | 0.81 | 0.29 | 0.91 | 0.32 | **0.96** | **0.44** |
| | tau2 | 0.94 | 0.39 | 0.89 | 0.29 | 0.89 | 0.33 | **0.99** | **0.44** |
| | tau3 | 0.91 | 0.38 | 0.86 | 0.29 | 0.90 | 0.33 | **0.96** | **0.44** |
| | tau4 | 0.94 | 0.40 | 0.89 | 0.30 | 0.91 | 0.33 | **0.98** | **0.44** |
| | Height | 0.99 | 1.53 | 0.65 | 1.23 | **0.96** | **1.36** | 0.78 | 1.49 |
| | Sex_1 | 0.92 | 1.74 | 0.78 | 1.53 | **0.95** | **1.78** | 0.92 | 1.92 |
| abalone | Sex_2 | **0.99** | **1.48** | 0.81 | 1.47 | 0.95 | 1.55 | 0.91 | 1.63 |
| | Shucked_weight | **0.96** | **1.40** | 0.68 | 1.23 | 0.92 | 1.48 | 0.82 | 1.36 |
| | intercept | 0.94 | 1.23 | 0.76 | 1.01 | **0.99** | **1.76** | 0.91 | 1.23 |

*Table 13.* Coverage (rate) and median width (size) over 100 repetitions of marginal credible intervals for logistic regression with real data. The method achieving at least the target coverage with the smallest size is shown in bold. If none of the methods achieve the target coverage, the method with the highest coverage is shown in bold. The target coverage is 0.95.

| Setup | Feature Name | TabMGP | | BB | | Copula | | Bayes | |
|---|---|---|---|---|---|---|---|---|---|
| | | Rate | Size | Rate | Size | Rate | Size | Rate | Size |
| rice | Eccentricity_Osmancik | 0.91 | 5.36 | 0.37 | 7.52 | **0.98** | **14.46** | 0.62 | 4.81 |
| | Extent_Osmancik | **0.99** | **1.19** | 0.93 | 2.11 | 1.00 | 4.14 | 0.93 | 1.40 |
| | Minor_Axis_Length_Osmancik | 0.93 | 3.91 | 0.49 | 5.55 | **0.97** | **11.15** | 0.65 | 3.38 |
| | intercept_Osmancik | 0.89 | 2.65 | 0.30 | 3.42 | **0.97** | **6.20** | 0.39 | 2.28 |
| sepsis | age_years_1 | 0.85 | 1.72 | **0.96** | **2.67** | 1.00 | 4.88 | 0.93 | 2.10 |
| | episode_number_1 | **1.00** | **1.19** | 0.93 | 2.54 | 1.00 | 3.28 | 0.99 | 1.83 |
| | intercept_1 | **1.00** | **1.77** | 0.98 | 2.91 | 1.00 | 6.42 | 0.97 | 1.95 |
| | sex_0male_1female_1_1 | **1.00** | **1.73** | 0.99 | 3.59 | 1.00 | 7.84 | 1.00 | 2.52 |
| banknote | V1_2 | 0.96 | 2.24 | 0.96 | 2.62 | 1.00 | 2.85 | **0.98** | **1.96** |
| | V2_2 | **0.97** | **2.23** | 0.98 | 2.49 | 0.97 | 2.44 | 0.94 | 1.71 |
| | V4_2 | **0.98** | **1.99** | 0.98 | 2.25 | 0.99 | 2.30 | 0.92 | 1.38 |
| | intercept_2 | 1.00 | 1.33 | 1.00 | 1.44 | 1.00 | 1.45 | **1.00** | **1.01** |
| mozilla | end_1 | 0.98 | 0.99 | 0.97 | 1.10 | 1.00 | 2.08 | **0.99** | **0.74** |
| | event_1 | 1.00 | 0.78 | 1.00 | 0.92 | 1.00 | 2.05 | **1.00** | **0.68** |
| | id_1 | 0.99 | 0.77 | 1.00 | 0.91 | 1.00 | 1.83 | **0.95** | **0.66** |
| | intercept_1 | 0.94 | 1.44 | 0.77 | 1.76 | **1.00** | **2.40** | 0.75 | 1.04 |
| | size_1 | 0.93 | 3.96 | 0.74 | 4.69 | **0.99** | **4.41** | 0.57 | 2.72 |
| skin | V1_2 | 1.00 | 3.00 | 1.00 | 3.29 | 1.00 | 3.23 | **0.99** | **1.59** |
| | V3_2 | 0.99 | 2.91 | 0.99 | 3.18 | 1.00 | 3.37 | **0.98** | **1.76** |
| | intercept_2 | **1.00** | **1.70** | 1.00 | 1.85 | 1.00 | 2.11 | 1.00 | 1.76 |
| blood | V1_2 | 0.82 | 1.80 | 0.91 | 2.64 | **0.98** | **4.18** | 0.83 | 1.63 |
| | V2_2 | 0.91 | 2.30 | 0.74 | 3.51 | **0.99** | **4.98** | 0.68 | 2.19 |
| | V4_2 | 0.81 | 1.76 | 0.79 | 2.74 | **1.00** | **4.01** | 0.77 | 1.91 |
| | intercept_2 | **1.00** | **1.48** | 0.93 | 2.14 | 1.00 | 3.68 | 0.92 | 1.28 |
| phoneme | V1_2 | **0.96** | **1.93** | 0.85 | 2.05 | 1.00 | 3.58 | 0.86 | 1.72 |
| | V2_2 | **0.95** | **1.84** | 0.87 | 2.15 | 0.98 | 3.44 | 0.80 | 1.46 |
| | V3_2 | 0.90 | 1.87 | 0.80 | 2.34 | **0.96** | **3.70** | 0.71 | 1.43 |
| | V4_2 | 0.91 | 1.68 | 0.80 | 1.97 | **0.95** | **3.68** | 0.75 | 1.24 |
| | V5_2 | 0.89 | 1.42 | 0.83 | 1.90 | **0.94** | **3.30** | 0.73 | 1.11 |
| | intercept_2 | **0.97** | **1.73** | 0.83 | 1.94 | 1.00 | 3.37 | 0.89 | 1.39 |
| telescope | fAlpha_h | 0.87 | 2.48 | 0.79 | 2.96 | **0.93** | **8.26** | 0.71 | 1.54 |
| | fAsym_h | **0.96** | **2.55** | 0.78 | 3.38 | 1.00 | 7.77 | 0.72 | 2.26 |
| | fConc1_h | **0.95** | **2.59** | 0.76 | 3.47 | 0.99 | 9.79 | 0.65 | 1.97 |
| | fDist_h | 0.93 | 1.99 | 0.81 | 2.66 | **0.98** | **6.56** | 0.72 | 1.60 |
| | fLength_h | 0.87 | 4.56 | 0.69 | 5.94 | **0.98** | **13.39** | 0.62 | 3.57 |
| | fM3Long_h | **0.95** | **2.44** | 0.87 | 3.45 | 0.99 | 7.24 | 0.80 | 2.22 |
| | fM3Trans_h | **0.97** | **2.21** | 0.88 | 3.12 | 1.00 | 6.63 | 0.81 | 1.92 |
| | fWidth_h | **0.95** | **4.38** | 0.70 | 5.57 | 0.98 | 11.94 | 0.67 | 3.62 |
| | intercept_h | **0.97** | **1.93** | 0.88 | 2.54 | 1.00 | 7.44 | 0.83 | 1.56 |

*Table 14.* Coverage (rate) and median width (size) over 100 repetitions of marginal credible intervals for multinomial logistic regression with the yeast dataset. The method achieving at least the target coverage with the smallest size is shown in bold. If none of the methods achieve the target coverage, the method with the highest coverage is shown in bold. The target coverage is 0.95.

| Setup | Feature Name | TabMGP | | BB | | Bayes | |
|---|---|---|---|---|---|---|---|
| | | Rate | Size | Rate | Size | Rate | Size |
| | alm_ME2 | **0.93** | **3.43** | 0.47 | 3.86 | 0.45 | 2.58 |
| | alm_ME3 | **0.95** | **3.16** | 0.56 | 4.22 | 0.70 | 2.81 |
| | alm_MIT | 0.85 | 1.20 | **0.89** | **1.49** | 0.71 | 1.03 |
| | alm_NUC | **0.93** | **0.79** | 0.90 | 0.97 | 0.90 | 0.79 |
| | gvh_ME2 | **0.94** | **2.39** | 0.56 | 3.27 | 0.57 | 2.19 |
| | gvh_ME3 | **0.94** | **1.41** | 0.82 | 1.95 | 0.73 | 1.49 |
| | gvh_MIT | **0.92** | **0.94** | 0.86 | 1.20 | 0.77 | 0.87 |
| | gvh_NUC | **0.94** | **0.61** | 0.93 | 0.86 | 0.86 | 0.64 |
| | intercept_ME2 | **0.89** | **6.35** | 0.39 | 10.10 | 0.43 | 4.42 |
| | intercept_ME3 | **0.95** | **3.53** | 0.54 | 4.87 | 0.67 | 3.01 |
| | intercept_MIT | **0.97** | **1.08** | 0.87 | 1.30 | 0.82 | 0.94 |
| | intercept_NUC | 1.00 | 0.70 | 1.00 | 0.81 | **1.00** | **0.62** |
| | mcg_ME2 | **0.90** | **2.74** | 0.61 | 4.14 | 0.59 | 2.47 |
| | mcg_ME3 | **0.92** | **1.63** | 0.82 | 2.11 | 0.81 | 1.64 |
| yeast | mcg_MIT | **0.90** | **0.90** | 0.90 | 1.12 | 0.78 | 0.85 |
| | mcg_NUC | **0.96** | **0.66** | 0.94 | 0.89 | 0.86 | 0.64 |
| | mit_ME2 | **0.93** | **2.48** | 0.57 | 2.93 | 0.62 | 2.62 |
| | mit_ME3 | **0.97** | **1.99** | 0.84 | 2.65 | 0.77 | 2.07 |
| | mit_MIT | **0.87** | **0.92** | 0.84 | 1.07 | 0.80 | 0.75 |
| | mit_NUC | **0.93** | **0.76** | 0.90 | 0.93 | 0.87 | 0.70 |
| | nuc_ME2 | **0.98** | **3.30** | 0.55 | 3.92 | 0.65 | 3.26 |
| | nuc_ME3 | **0.74** | **1.67** | 0.74 | 2.23 | 0.69 | 1.77 |
| | nuc_MIT | **0.94** | **1.34** | 0.83 | 1.61 | 0.76 | 1.28 |
| | nuc_NUC | **0.89** | **0.81** | 0.81 | 1.04 | 0.72 | 0.70 |
| | vac_ME2 | **0.99** | **2.06** | 0.63 | 2.62 | 0.71 | 2.31 |
| | vac_ME3 | **0.97** | **1.60** | 0.87 | 2.22 | 0.91 | 1.98 |
| | vac_MIT | **0.98** | **0.76** | 0.90 | 1.04 | 0.85 | 0.78 |
| | vac_NUC | **0.95** | **0.53** | 0.89 | 0.75 | 0.86 | 0.57 |

*Table 15.* Coverage (rate) and median width (size) over 100 repetitions of marginal credible intervals for multinomial logistic regression with the `wine` dataset. The method achieving at least the target coverage with the smallest size is shown in bold. If none of the methods achieve the target coverage, the method with the highest coverage is shown in bold. The target coverage is 0.95.

| Setup | Feature Name | TabMGP | | BB | | Bayes | |
|---|---|---|---|---|---|---|---|
| | | Rate | Size | Rate | Size | Rate | Size |
| | V10_3 | **0.97** | **1.25** | 0.75 | 2.30 | 0.73 | 1.65 |
| | V10_4 | **0.96** | **1.27** | 0.71 | 2.18 | 0.71 | 1.61 |
| | V10_5 | **0.96** | **1.34** | 0.74 | 2.25 | 0.70 | 1.69 |
| | V10_6 | **0.96** | **1.64** | 0.79 | 3.03 | 0.73 | 2.09 |
| | V11_3 | **0.93** | **1.94** | 0.77 | 2.72 | 0.76 | 2.09 |
| | V11_4 | **0.97** | **1.98** | 0.73 | 2.78 | 0.78 | 2.09 |
| | V11_5 | **0.93** | **2.16** | 0.71 | 2.93 | 0.78 | 2.21 |
| | V11_6 | **0.93** | **2.83** | 0.64 | 4.22 | 0.67 | 2.85 |
| | V1_3 | **0.97** | **1.21** | 0.72 | 2.22 | 0.74 | 1.50 |
| | V1_4 | **0.90** | **1.26** | 0.74 | 2.20 | 0.72 | 1.48 |
| | V1_5 | **0.88** | **1.35** | 0.74 | 2.33 | 0.68 | 1.59 |
| | V1_6 | **0.91** | **1.73** | 0.74 | 3.17 | 0.74 | 2.19 |
| | V2_3 | **0.90** | **1.12** | 0.79 | 1.97 | 0.74 | 1.29 |
| | V2_4 | **0.81** | **1.30** | 0.73 | 2.06 | 0.68 | 1.36 |
| | V2_5 | **0.82** | **1.45** | 0.69 | 2.23 | 0.70 | 1.51 |
| | V2_6 | **0.84** | **1.99** | 0.67 | 3.12 | 0.66 | 2.15 |
| | V3_3 | **0.96** | **1.14** | 0.76 | 2.16 | 0.65 | 1.49 |
| | V3_4 | **0.97** | **1.18** | 0.72 | 2.21 | 0.63 | 1.49 |
| | V3_5 | **0.99** | **1.37** | 0.80 | 2.33 | 0.76 | 1.66 |
| | V3_6 | **1.00** | **1.85** | 0.74 | 3.19 | 0.71 | 2.37 |
| wine | V4_3 | **0.98** | **1.49** | 0.70 | 2.60 | 0.74 | 1.94 |
| | V4_4 | **0.96** | **1.50** | 0.74 | 2.66 | 0.75 | 1.93 |
| | V4_5 | **0.93** | **1.65** | 0.72 | 2.69 | 0.75 | 2.04 |
| | V4_6 | **0.90** | **2.17** | 0.71 | 3.78 | 0.64 | 2.66 |
| | V5_3 | **0.98** | **1.33** | 0.68 | 2.46 | 0.74 | 1.74 |
| | V5_4 | **0.96** | **1.37** | 0.77 | 2.55 | 0.77 | 1.77 |
| | V5_5 | **0.94** | **1.98** | 0.82 | 2.90 | 0.81 | 2.22 |
| | V5_6 | **0.97** | **2.47** | 0.74 | 4.78 | 0.75 | 3.36 |
| | V6_3 | **0.79** | **1.77** | 0.69 | 3.10 | 0.56 | 1.75 |
| | V6_4 | **0.84** | **1.78** | 0.70 | 3.06 | 0.57 | 1.77 |
| | V6_5 | **0.84** | **1.92** | 0.72 | 3.19 | 0.60 | 1.85 |
| | V6_6 | **0.83** | **2.15** | 0.70 | 4.07 | 0.59 | 2.37 |
| | V9_3 | **0.97** | **1.29** | 0.75 | 2.50 | 0.64 | 1.72 |
| | V9_4 | **0.98** | **1.34** | 0.72 | 2.50 | 0.65 | 1.74 |
| | V9_5 | **0.95** | **1.46** | 0.71 | 2.57 | 0.65 | 1.82 |
| | V9_6 | **0.94** | **1.81** | 0.73 | 3.51 | 0.70 | 2.33 |
| | intercept_3 | **0.97** | **2.87** | 0.42 | 5.76 | 0.55 | 2.76 |
| | intercept_4 | **0.99** | **2.90** | 0.37 | 5.72 | 0.49 | 2.75 |
| | intercept_5 | **0.98** | **3.00** | 0.53 | 5.91 | 0.56 | 2.82 |
| | intercept_6 | **0.99** | **3.94** | 0.81 | 9.06 | 0.69 | 4.35 |

