# OpenReview forum: "TabMGP: Martingale Posterior with TabPFN"
_ICML.cc/2026/Conference — ICML 2026 regular_

### Official Review · Reviewer_pZiH · 2026-02-17

**Soundness:** 3
**Presentation:** 3
**Significance:** 3
**Originality:** 2
**Overall Recommendation:** 5
**Confidence:** 3

**Summary:**

The authors propose to use TabPFN for the prediction rule in Martingale Posteriors (MGPs). They introduce and explain MGPs and contrast their method with existing update-rules. In experiments, they show that the TabPFN update rule yields good uncertainty quantification in terms of  coverage and size of the credible set.

**Compliance With Llm Reviewing Policy:**

Affirmed.

**Final Justification:**

The authors successfully addressed my concerns in terms of the relationship to related work by Nagler and Rugamer. They also provided additional results regarding my concerns about some empirical findings, which addressed exactly the things I was worried about.

I think the paper opens a pathway into a very interesting direction for future research: using Tabular Foundation Models for constructing/improving statistical estimators/uncertainty quantification.

In the discussion with the authors I increased my score from 3 (weak reject) to 5 (accept).

**Key Questions For Authors:**

Please feel free to reply to my questions above. If my concerns regarding (a) convergence of \theta(F_N) and (b) advantages and discrepancies compared to Nagler and Rugamer can be resolved, I would vote for the paper to be accepted.

**Limitations:**

A more thorough discussion of the implications (theoretical, if possible) of non-convergence of the MGP would be helpful.

**Strengths And Weaknesses:**

Soundness:

As discussed by the authors, the TabPFN update-rule does not satisfy the martingale property. A more thorough discussion about the theoretical (negative) implications would be nice. What exactly would happen if no limiting distribution exists? What would it mean if, for example all \theta(F_N) converge only "approximately" to a limiting distribution up to some error?

Much more importantly, some of the experimental results seem to contradict Nagler and Rugamer:
- They claim to find that the limiting distribution implied by the TabPFN update rule does NOT converge (4.1. TabPFN Martingale Property Check in [1]). Could you repeat exactly their experiment to see if this is reproducible under their exact setup, and then what happens in your setup? (I think they don't do the Bayesian Bootstrap for their covariates)
- I would politely disagree that it is very obvious from your plots in Figure 6, Appendix F, that we see convergence across all the setups. Especially "skin", "yeast", "wine" really don't look like convergence is happening. Also the synthetic GMM cases. Could you perform rollouts with much larger N to see what is happening there? The L1-norm analysis in Figure 3 does look better, but this is just a simple sufficient condition for convergence.
In general, I find it problematic to check the behavior of a stochastic process VISUALLY at N=500 and then claim convergence.

Besides that, I like the experiments regarding the credible sets for the regression and classification setups. However, could you compare against Nagler and Rügamer here? Since they claim to perform better than using just TabPFN.

Presentation:

The paper is very well understandable and all concepts are explained well, presuming the reader is familiar with TabPFN. The flow of the paper is quite nice and everything is well-motivated. The figures are effective as well. Partially the writing feels slightly sloppy.

 A more thorough discussion of the work by Nagler and Rugamer, which looks very similar, is missing. Especially a discussion about what exactly the benefits and drawbacks of the presented method compared to Nagler and Rugamer [1] is, is missing

Significance:

The presented idea is simple, yet very meaningful. I appreciate that! Using tabular foundation models for the update-rule in MGPs is a very good idea. I can imagine that it leads to quite some more research in that direction.

Originality:

The work here is very similar to that of Nagler and Rugamer. A more detailed comparison, also in terms of experiments is missing.

Otherwise, the proposed combination of MGPs and TabPFN is very neat and original.


NIT: there is a typo around line 143 (RHS): "a one-step-ahead predictive distribution, ,"


References:

[1] https://arxiv.org/pdf/2505.11325

---

> ### Author Rebuttal · Authors · 2026-03-31
>
> We thank Reviewer pZiH for the positive assessment of the core idea and for explicitly stating that resolving (a) convergence and (b) the comparison with Nagler & Rügamer would lead to acceptance. We address both below.
>
>
> ### Convergence of $\theta(F_N)$
>
> **Martingale property vs. convergence of $\theta(F_N)$.** We find it helpful to
> draw a distinction between (1) whether the *predictive rule* strictly satisfies
> the martingale (or a.c.i.d.) property, and (2) whether the *functional*
> $\theta(F_N)$ converges. While the martingale property is a well-established
> *sufficient* condition for the convergence of $F_N$ (and therefore
> $\theta(F_N)$), it is not a *necessary* one.
>
> Nagler & Rügamer (NR) test the martingale condition on a Gamma(2,2)
> data-generating process (DGP) and find it is violated. Similarly, we tested the
> weaker a.c.i.d. condition on a logistic regression DGP (Appendix C) and found
> that TabPFN violates it too. Although evaluated on different DGPs, our
> conclusion aligns: TabPFN's predictive rule is not a strict martingale.
>
> We also replicated the experiment from NR's Section 4.1 (noting that some
> implementation details were omitted in their text) and present the results in
> [Figure 14](https://anonymous.4open.science/r/tabmgp-icml-rebuttal-D558/README.md). We
> observe that TabPFN's predictive rule does not deviate drastically from a
> martingale as observed by NR, particularly within the region supported by the
> data (around $y \in [0,10]$).
>
> Our core contribution explores what happens *after* this negative result. We
> demonstrate that despite this violation, the functional $\theta(F_N)$ still
> converges empirically, yielding posteriors with near-nominal coverage. Achieving
> sound empirical results while failing a sufficient theoretical condition is a
> significant finding, as it directly challenges the practical necessity of these
> strict constraints.
>
> **Visual convergence in Figure 6 and longer rollouts.** We acknowledge the
> concern that convergence is not visually obvious for the *skin*, *yeast*,
> *wine*, and GMM setups at $N = n + 500$. Therefore, we provide a longer rollouts
> for these setups in [Figure
> 13](https://anonymous.4open.science/r/tabmgp-icml-rebuttal-D558/README.md),
> demonstrating that the trajectories do indeed stabilize with a longer rollout.
>
> ### Comparison with Nagler & Rügamer
>
> **A note on versions.** All statements regarding NR refer to arXiv:2505.11325v2
> (available at the time of our January 2026 submission). We are happy to discuss
> later versions during the discussion period.
>
> **Methodological distinction.** While both NR's method and ours utilize TabPFN within the MGP framework, they differ fundamentally:
>
> 1. **Mechanism:** NR use TabPFN solely to *initialize* the predictive
>    distribution and rely on the Gaussian copula for forward sampling. We retain
>    TabPFN as the predictive rule *throughout*.
> 2. **Theory:** NR achieve martingale validity by copula construction; we rely on
>    empirical validation under weaker-than-martingale conditions.
> 3. **Target:** NR focus on conditional inference; we focus on posterior
>    inference for loss-defined functionals $\theta(F_\infty)$.
> 4. **Tuning:** NR require copula bandwidth tuning, which we found numerically
>    unstable. TabMGP requires no such tuning beyond the rollout length $N$
>    required for all MGP methods.
>
> We provide a direct experimental comparison in [Table
> 14](https://anonymous.4open.science/r/tabmgp-icml-rebuttal-D558/README.md).
> Because NR's method does not apply off-the-shelf to our setting (they target
> pointwise inference at a fixed $x^\star$ rather than global parameters), we
> adapted their key idea (Copula initialized with TabPFN) for a fair comparison.
>
> Since the forward rule is identical in both cases, we saw minimal differences
> between NR's adapted method and the original copula baseline across most setups.
> The copula methods generally perform better in simpler synthetic regression
> setups, where the Gaussian copula update has a structural advantage, whereas
> TabMGP is preferred in most real-world setups. We note that NR predominantly
> evaluate their method on synthetic regression experiments, a regime where we
> agree copula methods excel.
>
> **Theoretical implications of non-convergence.** If $F_N$ fails to converge,
> $\theta(F_N)$ may oscillate or diverge, rendering the MGP posterior—defined as
> the distribution of $\theta(F_\infty)$—ill-defined. Even when $\theta(F_\infty)$
> does exist, practical implementation requires stopping at a finite $N$, meaning
> the quality of the approximation hinges on the proximity of $\theta(F_N)$ to
> $\theta(F_\infty)$. Our path stability diagnostic (Figure 3) is designed to
> detect this: if the $L_1$ trajectory has not plateaued, practitioners should
> increase $N$. We will expand upon this discussion in the revision.

---

> > ### Author Rebuttal · Reviewer_pZiH · 2026-04-01
> >
> > Thank you for your response!
> >
> > I agree with the authors that the longer rollouts provide clearer evidence for convergence.
> >
> > My concerns regarding the differences/discrepancies compared to Nagler and Rugamer have also been addressed with the discussion here and the additional experimental results.
> >
> > Furthermore, I think that it is important to give somewhat negative or theoretically surprising findings, such as the finding that the Martingale Property does not hold for the TabPFN rule appropriate attention, but can still be used for effective inference.
> >
> > I have adjusted my score accordingly.

---

### Official Review · Reviewer_zQs6 · 2026-03-12

**Soundness:** 3
**Presentation:** 3
**Significance:** 3
**Originality:** 3
**Overall Recommendation:** 4
**Confidence:** 4

**Summary:**

This paper introduces TabMGP, an MGP built on TabPFN for tabular uncertainty quantification. It treats TabPFN as a predictive rule to forward-simulate continuations of the dataset; across rollouts, it computes a target functional, such as regression parameters defined as minimizers of an expected loss, and uses the resulting empirical distribution as a posterior uncertainty estimate. For a loss $\ell(z,\theta)$ and data-generating distribution $F$, the target takes the form $\theta(F) \in \arg\min_{\vartheta} \int \ell(z,\vartheta)\, dF(z)$. Since it's unclear whether TabPFN satisfies the standard sufficient conditions in martingale posterior theory, the paper relies on empirical validation, using diagnostics based on path stability over the forward horizon and frequentist coverage of credible sets. Numerical experiments across a broad collection of synthetic and real-data settings compare TabMGP with several alternative MGP constructions and Bayesian/asymptotic baselines, and demonstrate that TabMGP often achieves good coverage with relatively tight uncertainty sets.

**Compliance With Llm Reviewing Policy:**

Affirmed.

**Final Justification:**

My concerns have been adequately addressed, and I will maintain my current score.

**Key Questions For Authors:**

1. In practice, how do you choose the forward horizon and how sensitive are the results to this choice?

2. Since the posterior samples can be non-Gaussian, do your main findings still hold with a more flexible credible set than an ellipsoidal one?

3. What failure cases have you seen (e.g., instability or poor calibration), and can your diagnostics detect them reliably?

**Limitations:**

The limitations discussion could be stronger. In particular, it will be helpful to state that the evidence is mainly empirical, and outline likely failure cases as well as practical checks, especially under distribution shift.

**Strengths And Weaknesses:**

**Strengths**

- The paper is practical in the sense that it uses TabPFN to provide uncertainty for downstream targets without building a full generative model.

- The authors are honest about the lack of theoretical guarantees, and they support the approach with empirical checks.

- The experiments are broad and the appendix includes enough implementation detail to reproduce.


**Weaknesses**

- The evidence is mainly empirical. It is still unclear when the method works or fails, and how to choose the forward horizon.

- The credible sets are simple, but the posterior samples can be non-Gaussian. It is unclear if the conclusions hold with other uncertainty sets.

---

> ### Author Rebuttal · Authors · 2026-03-31
>
> We thank Reviewer zQs6 for recognizing that the paper is "practical in the sense that it uses TabPFN to provide uncertainty for downstream targets without building a full generative model," that we are "honest about the lack of theoretical guarantees," and that the "experiments are broad and the appendix includes enough implementation detail to reproduce." We address your questions below.
>
> **Forward horizon and sensitivity (Q1).** For TabMGP, we use $N = n + 500$
> forward-sampling steps (i.e., 500 steps beyond the observed data), whereas BB
> and Copula use 2000 steps. The choice for TabMGP is guided by the path stability
> diagnostic (Figure 3): we monitor the scaled $L_1$-norm between $\theta(F_n)$
> and $\theta(F_N)$, selecting an $N$ large enough for the trajectories to
> stabilize. In practice, we recommend running this diagnostic and increasing $N$
> until the $L_1$ trajectory plateaus (or until computational resources are
> exhausted).
>
> We acknowledge that a more systematic stopping criterion would be valuable. In [Table 13](https://anonymous.4open.science/r/tabmgp-icml-rebuttal-D558/README.md), we report the coverage of TabMGP at various rollout lengths for selected setups where convergence is slower (skin, yeast, wine, GMM). We observe that the coverage—and, to a large extent, the credible set size—stabilizes after 500 rollouts.
> In our response to Reviewer pZiH, we also provide an $L_1$-convergence plot with a longer rollouts ($N = n + 1000$) in [Figure 13](https://anonymous.4open.science/r/tabmgp-icml-rebuttal-D558/README.md). These results directly address the method's sensitivity to the choice of $N$.
>
>
> **Non-Gaussian posteriors and flexible credible sets (Q2).** This is a valuable
> point, though the challenge of constructing high-dimensional credible sets from
> samples applies broadly to any sampling-based method (including Bayes via NUTS).
> We opted for the ellipsoidal approximation because reliably estimating highest
> posterior density (HPD) sets in high dimensions requires substantially more than
> the 100 posterior samples per setup generated in our experiments.
>
> To provide an alternative perspective that does not rely on an ellipsoidal
> shape, we also report the marginal credible interval coverages in Appendix H.
> These marginal results largely align with our joint coverage findings: TabMGP
> consistently provides near-nominal coverage and meaningful uncertainty across
> many real-world setups.
>
> **Failure cases and diagnostics (Q3).** We have observed two main failure modes:
>
> 1. **Classification with low $n/p$:** In the synthetic logistic regression
>    setups (Table 2), TabMGP can exhibit undercoverage. This is likely because
>    $n=100$ context observations are insufficient for TabPFN to produce
>    well-calibrated predictive distributions in these settings.
>
> 2. **Slow convergence:** For some real-world datasets with complex structures
>    (e.g., *wine*, *yeast*), the path stability diagnostic indicates slower
>    convergence, requiring a larger $N$. While this is not divergence—the
>    trajectories remain bounded without systematic drift—convergence was not
>    fully reached within the global $N = 500$ used across all datasets. As
>    detailed in our response to Reviewer pZiH ("Convergence"), longer rollouts
>    confirm that these paths eventually stabilize.
>
> Our diagnostics—path stability (Figures 3 and 6), posterior concentration
> (Figure 2) and the coverage itself-can detect these issues. The path stability
> check is the most direct indicator: if the $L_1$ trajectory has not plateaued,
> practitioners should increase $N$ or interpret the results with caution.
>
> Following your suggestion, we will strengthen the limitations section in the
> revision to explicitly state that our evidence is primarily empirical, outline
> these failure modes, and detail the necessary practical checks.

---

> > ### Author Rebuttal · Reviewer_zQs6 · 2026-04-03
> >
> > Thank you for the response. My main concerns have been adequately addressed. In particular, the authors have clarified the practical choice of the forward horizon, given additional discussion of the non-Gaussian posterior issue and identified the main failure cases and diagnostics. I have no further questions.

---

### Official Review · Reviewer_J8nR · 2026-03-16

**Soundness:** 2
**Presentation:** 4
**Significance:** 2
**Originality:** 3
**Overall Recommendation:** 4
**Confidence:** 4

**Summary:**

The TabPFN predictive is used within the martingale posterior framework by Fong et al. 2023 to yield a martingale posterior over the population parameters $\theta$, assumed in the experiments to be the coefficients of linear regression or logistic regression. While the TabPFN predictive departs from the sufficient conditions for the existence of limiting distribution $F_\infty$ that were considered in previous work, such as the martingale condition, the resulting martingale posterior demonstrates superior coverage compared to standard Bayesian posteriors and martingale posteriors with hand-crafted predictives.

**Compliance With Llm Reviewing Policy:**

Affirmed.

**Final Justification:**

The rebuttal addressed my main concerns.

**Key Questions For Authors:**

1. Could the authors please comment on the practical use cases of the target parameters considered in this supervised setting? What are other types of target parameters in classification or regression for which obtaining the posterior would be interesting? Especially helpful would be a case study inspecting the TabMGP for coefficients associated with easy-to-understand covariates for one of the real-world setups.
2. The authors resort to the urn scheme for forward sampling of the covariates, as they tend to be high-dimensional. But this can be a problem for small $n$ settings. If we had access to one-step-ahead predictives for the covariates from a separate modeling effort, how do we expect the conditional predictives of TabPFN and the resulting TabMGP to respond? Is it possible that the joint one-step-ahead predictives over $(X, Y)$ will be more martingale?

**Limitations:**

Yes

**Strengths And Weaknesses:**

### Soundness
- So far as TabMGP competes with standard Bayes requiring prior + likelihood specification, I believe the "standard Bayes posterior" baseline should include multiple priors of varying degrees of informativeness. Currently, only the asymptotic approximation of the posterior is used as the prior. Including more diffuse priors in the comparison will help understand what "effective prior" is implied by the TabMGP predictive.
- For real-data experiments, the "ground-truth" $\theta$ is simply taken to be $\theta$ computed from the entire dataset. As a middle ground between simulated and real-data setups, to aim at complexity in the $x \mapsto y$ mapping while also having access to true $\theta$, we can draw parameters of a reasonably small MLP that takes as input $x$ and outputs $y$. The $x$ can be taken from a real dataset.
- Path stability check: Figure 3 should instead show the average across realizations. Currently, Figure 3 gives the incorrect impression that the $L_1$-norm between $\theta(F_N)$ and $\theta(F_n)$ converge to a value almost monotonically, while the individual realizations and their averages (thick solid lines) in Figure 6 are more consistent with paths that recover from intermediate spikes, for many setups. If I'm interpreting the captions correctly, the difference seems to be that Figure 3 is showing a single realization of $z_{1:n}$, which may not be representative of other realizations. Including potential explanations of intermediate drift (e.g., extreme TabPFN predictive sample at a given $n$) would be helpful as well.
- Shape of the posteriors: I don't think we can conclude from the skewness and multimodal structure of TabMGP in Figure 4 that TabMGP is more expressive. These features could be artifacts due to intermediate drifts at finite $n$ caused by outlier predictive samples from the TabPFN. In general, these features can result from unidentifiability of the parameter, and it's possible the Bayes baseline lacks them because its prior (the asymptotic posterior) is quite tight and unimodal, and there isn't sufficient data ($n=100$ for concrete) to depart from the prior shape. Also, the copula baseline seems to share skewness and multimodality, though more diffuse. It would also be helpful to show the equivalent of Figure 4 for other parameters, for other setups.

### Presentation
The manuscript is clearly written and well structured. The implementation is easy to understand. The Related Work section is particularly comprehensive, and I appreciated explicit references to recent and concurrent work.
- Each setup can be shown in a different color or line style for both Figures 3 and 6, to enable easy comparison between the figures. It would also be helpful if the regression and classification setups in Figure 6 can be shown separately, and the real setups ordered by $n/p$ within each.
- Typo: double comma in Sec 3.1, second paragraph
- NUTS algorithm not cited in the main text

### Significance
My biggest concern about this work is that the target parameters $\theta$ considered, which are coefficients of linear or logistic regression models, are unlikely to be of real practical interest. For linear regression, the $\theta$ are $d+1$-dimensional, where $d$ is the dimensionality of $x$ and, for logistic regression, $\theta$ are $K \times (d+1)$-dimensional. Particularly when $d$ is large, which is the setting the authors focus on, the $\theta$ is not easy to interpret and the $\theta$ posterior is not the end goal. More relevant inference scenarios might be when there is a misspecified *yet interpretable* model, such as a heavily regularized linear model, with few parameters that can be inspected by a domain expert. Unless it is meaningful to reason about $\theta$, it would be enough to work with TabPFN predictions and not necessary to obtain the TabMGP.

### Originality
This work represents an interesting and novel combination of powerful predictives from transformer-based foundation models and the growing literature on prediction-oriented Bayes. As the authors state, the strong empirical performance of TabMGP suggest that there is room for theory to catch up; the cid or acid properties remain sufficient conditions and it would be compelling to work out less restrictive conditions more relevant for powerful modern predictives.

---

> ### Author Rebuttal · Authors · 2026-03-31
>
> We thank Reviewer J8nR for their expert and thorough engagement with our work. We are grateful for the recognition that the "manuscript is clearly written and well structured," that the "implementation is easy to understand," and that the "Related Work section is particularly comprehensive." We take the critical questions seriously and address each below.
>
>
> **Multiple priors for the Bayes baseline (Soundness).** We appreciate this
> suggestion. We originally evaluated the Bayes baseline with a diffuse prior but
> excluded it because it yielded poor results (excessively wide credible sets with
> inflated coverage), which we felt unfairly penalized the baseline. Using an
> empirical Bayes prior—an asymptotic approximation centered at the maximum
> likelihood estimate—was our best attempt to give the Bayes baseline a strong
> advantage by using the data to inform the prior.
>
> We have included the results using a diffuse prior in [Table
> 14](https://anonymous.4open.science/r/tabmgp-icml-rebuttal-D558/README.md),
> which demonstrates that the Bayes posterior with a diffuse prior consistently
> produces exceedingly wide credible sets.
>
>
> **Semi-synthetic MLP experiment (Soundness).** We appreciate this suggestion.
> However, we feel that the experimental design is cleaner when the two regimes
> are kept separate: fully synthetic setups where the ground truth
> $\theta(F^\star)$ is known exactly, and fully real-data setups where it is not.
> A semi-synthetic MLP setup introduces a new source of ambiguity—the MLP
> parameters are "true" only in the sense that they were drawn, not because they
> reflect any real data-generating process—and it is unclear what additional
> insight this provides beyond what our existing synthetic setups already offer.
> That said, we acknowledge this is an interesting direction worth pursuing in
> future work, particularly as the community develops richer benchmarks for
> uncertainty quantification methods.
>
> **Figure 3 Clarification (Soundness).** Each trajectory in Figure 3 is
> intentionally based on a single realization of $z_{1:n}$. The convergence
> diagnostic evaluates whether $\theta(F_N)$ stabilizes as $N$ grows *conditional
> on a given observed dataset*—reflecting the practical reality that an analyst
> only has access to one $z_{1:n}$. Averaging across multiple draws of $z_{1:n}$
> would conflate this convergence check with a frequentist evaluation across
> datasets, which our coverage tables already address.
>
> Figure 3 serves as a compact summary, overlaying all 30 setups so readers can
> assess the overall behavior at a glance. Figure 6 (Appendix F) provides a more
> granular view, showing the individual rollouts (grey lines) and their average
> (black solid line) for each setup. Thus, Figure 3 simply compiles the black
> solid lines from each panel of Figure 6 into a single plot.
>
> **Shape of the Posteriors / Figure 4 (Soundness).** We agree that the word
> "suggests" was too strong, and we will soften this claim in the revision. We
> also note that density plots for all parameters across multiple setups are
> already available in Appendix G (Figures 7–11), which provide a broader view of
> these posterior shapes.
>
> **Practical significance of target parameters (Key Question 1).** We agree with
> the reviewer that using our method to fit a misspecified but interpretable model
> is a compelling use case. While we do not feature a standalone case study in the
> main text, our experiments already include real-world setups where the target
> parameters are directly interpretable coefficients. For instance, the "abalone"
> setup examines the determinants of abalone age. In this setting, the posterior
> is computed for coefficients corresponding to easy-to-understand covariates: the
> abalone's height, sex, and shucked weight. The marginal coverage for these
> interpretable coefficients is detailed at the bottom of Table 9, and their
> corresponding posterior density plots are visualized in Figure 9.
>
> **Urn scheme for covariates at small $n$ (Key Question 2).** We agree that the
> Bayesian bootstrap for covariates is a coarse approximation in the small-$n$
> regime. Incorporating a richer generative model for the covariates could improve
> empirical performance. However, it is unclear if this would yield joint $(X, Y)$
> predictives that more strictly satisfy the martingale property, as the condition
> applies to the joint predictive rule rather than just the marginals.
> Nonetheless, we consider this a compelling direction for future work.

---

> > ### Author Rebuttal · Reviewer_J8nR · 2026-04-02
> >
> > Thank you for your responses.
> >
> > > included the results using a diffuse prior in Table 14, which demonstrates that the Bayes posterior with a diffuse prior consistently produces exceedingly wide credible sets
> >
> > Thank you for including this experiment.
> >
> > > unclear what additional insight this provides beyond what our existing synthetic setups already offer
> >
> > It would capture more complex mappings than the synthetic setups, currently only covering linear and logistic mappings, while having a known ground truth. The covariate distribution would also be from real data. An alternative is to simulate DAGs as TabPFN does for training.
> >
> > > Averaging across multiple draws of $z_{1:n}$ would conflate this convergence check with a frequentist evaluation across datasets
> >
> > I believe there's been a misunderstanding. My suggestion for Figure 3 was not to average the curves *across* datasets, but to show a representative summary-curve for *each* dataset, for example by replacing each trajectory with the corresponding black curve (the average across realizations) in Appendix Figure 6. Currently Figure 3 shows a single, handpicked realization from each dataset and this gives the false impression that all trajectories tend to converge near-monotonically, when in fact, Appendix Figure 6 reveals that many trajectories seem to carry intermediate spikes.

---

> > > ### Author Response · Authors · 2026-04-06
> > >
> > > Thank you for your continue engagement.
> > >
> > > **Semi-synthetic MLP experiment**
> > >
> > > We understand the reviewer suggested using a semi-synthetic setup where the $x$ values come from a real dataset, while $y$ is generated via a synthetic MLP mapping $x \mapsto y$. This ensures we have access to the true data-generating process (DGP) to compute the true $\theta$. We conducted these experiments using two real datasets: one with a continuous response (Concrete) and one with a binary response (Phoneme). We trained a small MLP on these datasets and used it to generate fresh observations $y_{1:n}$ (given the real $x_{1:n}$) for our coverage experiments.
> > >
> > > The table below presents the coverage of the joint 95% credible sets. We will include these results in the revised manuscript.
> > >
> > > | Setup | TabMGP (Rate) | TabMGP (Size) | BB (Rate) | BB (Size) | Copula (Rate) | Copula (Size) | Bayes (Rate) | Bayes (Size) | Asymptotics (Rate) | Asymptotics (Size) |
> > > | :--- | :---: | :---: | :---: | :---: | :---: | :---: | :---: | :---: | :---: | :---: |
> > > | concrete-semi | 0.86 | 0.1465 | 0.84 | 0.1620 | 1.00 | 0.3580 | 0.94 | 0.2518 | 1.00 | 0.5189 |
> > > | phoneme-semi | 0.98 | 4.1763 | 0.94 | 5.1099 | 1.00 | 10.5623 | 0.76 | 1.5573 | 1.00 | 3.1998 |
> > >
> > > We observe that TabMGP and BB achieve close to the ideal 95% coverage, with TabMGP producing smaller set sizes. Copula consistently overcovers with an excessively wide set. Bayes performs well on Concrete but severely undercovers on Phoneme. In contrast, the asymptotic approximation performs well on Phoneme (the smallest set among methods with a close-to-nominal coverage) but produces a very wide set for Concrete. We hope this serves as further evidence that TabMGP performs competitively across diverse settings, without requiring manual model-prior specification or hyperparameter tuning.
> > >
> > >
> > > **Convergence plot over realizations**
> > >
> > > We suspect the confusion may stem from the slightly non-standard nomenclature used in our work. Here, we use *dataset* to refer to a single realization $z_{1:n}$ from the data-generating process (DGP). For our coverage experiments in Tables 1 and 2, we draw 100 datasets/realizations of $z_{1:n}$ from each DGP to compute the coverage.
> > >
> > > Figure 3 indeed corresponds to the black curves in Figure 6, as requested by the reviewer, and is *not* handpicked from one of the grey lines in Figure 6. Each grey curve in Figure 6 corresponds to a rollout $\{z_{1:n}, Z_{n+1:N}\}$, and the black curve represents the average over these rollouts (marginalizing out the randomness in $Z_{n+1:N}$). All trajectories in Figures 3 and 6 (both black and grey) are computed from the same fixed $z_{1:n}$, which was also *not* handpicked (the realization of $z_{1:n}$ was generated with seed 1001). We compute the convergence plot only for a single realisation of $z_{1:n}$, since in practice the analyst observes only one such realisation from the unknown true DGP.
> > >
> > > Nevertheless, we understand the reviewer might still be concerned about convergence across different realizations of $z_{1:n}$ from the same DGP. Therefore, we have generated additional convergence plots using 20 realizations of $z_{1:n}$ across all 30 DGPs/setups, presented in [Figure 15](https://anonymous.4open.science/r/tabmgp-icml-rebuttal-D558/README.md). Each subplot contains 20 trajectories, one for each realization. We observe that the trajectories in Figure 15 generally converge, except for a few that spike or diverge. These edge cases typically correspond to realizations that lack diversity; as a result, TabPFN tends to produce predictions that lead to an ill-conditioned design matrix in the loss function (resulting in an unstable $\theta$). We will emphasize these failure cases alongside the new findings in the revised manuscript.
> > >
> > >
> > > We hope these additional experiments have adequately addressed any remaining concerns.

---

### Decision · Program_Chairs · 2026-04-30

**Decision:**

Accept (regular)

**Comment:**

The martingale posterior framework (Fong et al., 2023) produces Bayesian posterior samples by forward-sampling future data points from a predictive rule and computing a statistic of interest on each completed dataset. This paper plugs a pre-trained tabular transformer (TabPFN) in as the predictive rule, avoiding prior/likelihood specification and MCMC. The procedure itself — autoregressive sampling from TabPFN, then solving for regression coefficients—is straightforward–so the contribution is mostly empirical: showing that this works despite TabPFN violating the martingale property that the framework's theory requires. Experiments cover linear and logistic regression coefficients across 11 synthetic DGPs and 19 UCI datasets.

Reviewers found the idea simple and well-validated, with near-nominal frequentist coverage across 30 settings (J8nR, pZiH, zQs6). The martingale violation finding was itself seen as a contribution — it suggests the MGP framework is more robust than its theoretical assumptions require (pZiH).

pZiH raised a main concern: concurrent work by Nagler & Rugamer (2025) uses TabPFN similarly but switches to a parametric copula to preserve martingale validity. The authors drew a clear distinction — TabMGP retains TabPFN throughout, arguing the martingale property is unnecessary. J8nR noted that on some datasets the posterior estimates required more simulated points to stabilize; the authors demonstrated this with longer rollouts.

Oddly, no reviewer raised computational cost, but TabMGP is clearly computationally much more expensive: 70–200 seconds per posterior sample (uncompiled PyTorch) compared to under 60 seconds total for NUTS-based MCMC (compiled JAX). For 1000 posterior samples this gap is substantial, and so the authors have yet to show a practical advantage over simply running MCMC with a reasonable prior. The experimental scope is also narrow — only regression coefficients on small datasets (n ≤ 100), though the framework is in principle more general.  The authors may wish to check out Efficient Autoregressive Inference for Transformer Probabilistic Models by C Hassan, N Loka, et al. (ICLR 2026). This could speed up the sampling substantially.

All three reviewers moved toward acceptance (scores 3→4, 4→4, 3→5). The paper demonstrates a useful proof of concept — pretrained models can serve as MGP predictive rules to produce well-calibrated posterior samples — but the computational cost and narrow experimental scope temper the practical impact. I recommend weak accept.